# Why Do Deep Residual Networks Generalize Better than Deep Feedforward Networks? — A Neural Tangent Kernel Perspective

**Kaixuan Huang**[*]
Peking University
hackyhuang@pku.edu.cn

**Yuqing Wang**[*]
Georgia Institute of Technology
ywang3398@gatech.edu

**Molei Tao**
Georgia Institute of Technology
mtao@gatech.edu

**Tuo Zhao**
Georgia Institute of Technology
tourzhao@gatech.edu

## Abstract

Deep residual networks (ResNets) have demonstrated better generalization performance than deep feedforward networks (FFNets). However, the theory behind such a phenomenon is still largely unknown. This paper studies this fundamental problem in deep learning from a so-called "neural tangent kernel" perspective. Specifically, we first show that under proper conditions, as the width goes to infinity, training deep ResNets can be viewed as learning reproducing kernel functions with some kernel function. We then compare the kernel of deep ResNets with that of deep FFNets and discover that the class of functions induced by the kernel of FFNets is asymptotically not learnable, as the depth goes to infinity. In contrast, the class of functions induced by the kernel of ResNets does not exhibit such degeneracy. Our discovery partially justifies the advantages of deep ResNets over deep FFNets in generalization abilities. Numerical results are provided to support our claim.

## 1 Introduction

Deep Neural Networks (DNNs) have made significant progress in a variety of real-world applications, such as computer vision [1, 2, 3], speech recognition, natural language processing [4, 5, 6], recommendation systems, etc. Among various network architectures, Residual Networks (ResNets, [7]) are undoubtedly a breakthrough. Residual Networks are equipped with residual connections, which skip layers in the forward step. Similar ideas based on gating mechanisms are also adopted in Highway Networks [8], and further inspire many follow-up works such as Densely Connected Networks [9].

Compared with conventional Feedforward Networks (FFNets), residual networks demonstrate surprising generalization abilities. Existing literature rarely considers deep feedforward networks with more than 30 layers. This is because many experimental results have suggested that very deep feedforward networks yield worse generalization performance than their shallow counterparts [7]. In contrast, we can train residual networks with hundreds of layers, and achieve better generalization performance than that of feedforward networks. For example, ResNet-152 [7], achieving a $19.38\%$ top-1 error on the ImageNet data set, consists of 152 layers; ResNet-1001 [10], achieving a $4.92\%$ error on the CIFAR-10 data set, consists of 1000 layers.

Despite the great success and popularity of the residual networks, the reason why they generalize so well is still largely unknown. There have been several lines of research attempting to demystify this phenomenon. One line of research focuses on empirical studies of residual networks, and provides

---

[*]Equal contribution.

intriguing observations. For example, [11] show that residual networks behave like an ensemble of weakly dependent networks of much smaller sizes, and meanwhile, they also show that the gradient vanishing issue is also significantly mitigated due to these smaller networks. [12] further provide a more refined elaboration on the gradient vanishing issue. They demonstrate that the gradient magnitude in residual networks only shows sublinear decay (with respect to the layer), which is much slower than the exponential decay of gradient magnitude in feedforward neural networks. [13] propose a visualization approach for analyzing the landscape of neural networks, and further demonstrate that residual networks have smoother optimization landscape due to the skip-layer connections.

Another line of research focuses on theoretical investigations of residual networks under *simplified network architectures*. A commonly adopted structure, which is a reformulation of FFNets, is

$$x_\ell = \phi(x_{\ell-1} + \alpha W_\ell x_{\ell-1}), \tag{1}$$

where $\ell$ is the number of layers and the skip-connection only bypasses the weight matrix $W_\ell$ at each layer [14, 15, 16, 17, 18]. Specifically, [16] study the optimization landscape with linear activation; [17] study using Stochastic Gradient Descent (SGD) to train a two-layer ResNet. [18] study using Gradient Descent (GD) to train a two-layer non-overlapping residual network. [14, 15] both take the perturbation analysis approach to show convergence of such ResNets. A more realistic structure is

$$x_\ell = x_{\ell-1} + \phi(\alpha W_\ell x_{\ell-1}), \tag{2}$$

where the skip-connection bypasses the activation function [19, 20]. [20] only consider separable setting and take the perturbation analysis to show the convergence and generalization property of such ResNet. These results, however, are only loosely related to the generalization abilities of residual networks, and often considered to be overoptimistic, due to the oversimplified assumptions.

Some more recent works provide a new theoretical framework for analyzing *overparameterized* neural networks [21, 22, 23, 24, 14, 25, 26, 27]. They focus on connecting two- or three-layer over-parameterized (sufficiently wide) neural networks to *reproducing kernel Hilbert spaces*. Specifically, they show that under proper conditions, the weight matrices of a well trained overparameterized neural network (achieving any given small training error) are actually very close to their initialization. Accordingly, the training process can be described as searching within some class of reproducing kernel functions, where the associated kernel is called the "*neural tangent kernel*" (*NTK*, [21]) and only depends on the initialization of the weights. Accordingly, the generalization properties of the overparameterized neural network are equivalent to those of the associated NTK function class. Based on such a framework, [19] derived the NTK of the ResNet (2) when only the last layer is trained, and proved the convergence of such ResNet. However, they did not provide an explicit formula for the NTK when all layers are trained, which is required for characterizing the generalization property of ResNets.

To better understand the generalization abilities of deep feedforward and residual networks, we propose to investigate the NTKs associated with these networks when all but the last layers are trained, and consider the case when both widths and depths go to infinity[2]. For the structure of ResNets, we adopt (2) only with a slight modification, since it captures the essence of the skip-connection; see Section 2

$$x_\ell = x_{\ell-1} + \alpha \sqrt{\frac{1}{m}} V_\ell \sigma_0 \Big( \sqrt{\frac{2}{m}} W_\ell x_{\ell-1} \Big). \tag{3}$$

Specifically, we prove that similar to what has been shown for feedforward networks [21], as the width of deep residual networks increases to infinity, training residual networks can also be viewed as learning reproducing kernel functions with some NTK. However, such an NTK associated with the residual networks exhibits a very different behavior from that of feedforward networks.

To demonstrate such a difference, we further consider the regime, where the depths of both feedforward and residual networks are allowed to increase to infinity. Accordingly, both NTKs associated with deep feedforward and residual networks converge to their limiting forms sublinearly (in terms of the depth). For notational simplicity, we refer to the limiting form of the NTKs as the limiting NTK. Besides asymptotic analysis, we also provide nonasymptotic bounds, which demonstrate equivalence between limiting NTKs and neural networks with sufficient depth and width.

When comparing their limiting NTKs, we find that the class of functions induced by the limiting NTKs associated with deep feedforward networks is essentially not learnable. Such a class of

functions is sufficient to overfit training data. Given any finite sample size, however, the learned function cannot generalize. In contrast, the class of functions induced by the limiting NTKs associated with deep residual networks does not exhibit such degeneracy. Our discovery partially justifies the advantages of deep residual networks over deep feedforward networks in terms of generalization abilities. Numerical results are provided to support our claim.

Our work is closely related to [28]. They also investigate the so-called "Gaussian Process" kernel induced by feedforward networks under the regime where the depth is allowed to increase to infinity. However, their studied neural networks are essentially some specific implementations of the reproducing kernels using random features, since the training process only updates the last layer of the neural networks, and keeps other layers unchanged. In contrast, we assume the training process updates all layers except for the last layer.

**Notations**: We use $\sigma_0(z) = \max(0, z)$ to denote the ReLU activation function in neural networks. We use $\sigma(z)$ to denote the normalized ReLU function $\sigma(z) = \sqrt{2}\max(0, z)$. The derivative [3] of ReLU function (step function) is $\sigma_0'(z) = \mathbb{I}_{\{z \geq 0\}}$. Then $\sigma'(z) = \sqrt{2}\mathbb{I}_{\{z \geq 0\}}$ is the normalized step function. We use $D$ to denote the input dimension and $\mathbb{S}^{D-1}$ to denote the unit sphere in $\mathbb{R}^D$. We use $m$ to denote the network width (the number of neurons at each layer) and $L$ to denote the depth. Let $\mathcal{M}_+^2$ be the set of all $2 \times 2$ positive semi-definite matrices. We use $\mathcal{F}$ to denote the set of all symmetric and positive semi-definite functions from $\mathbb{R}^D \times \mathbb{R}^D$ to $\mathbb{R}$. We use $\|\cdot\|_{\max}$ to denote the entry-wise $\ell_\infty$ norm for matrices and use $\|\cdot\|$ to denote the $\ell_2$ norm for vectors and the spectral norm for matrices. We use $\mathrm{diag}(\cdot)$ to denote the diagonal matrix. We use $I_n$ to denote the $n \times n$ identity matrix. We use $x$ and $\tilde{x}$ to denote a pair of inputs. We use $x_\ell$ and $\tilde{x}_\ell$ to denote the output of the $\ell$-th layer of a network for the input $x$ and $\tilde{x}$, respectively. We use $f$ and $\tilde{f}$ to denote the final output of the network for $x$ and $\tilde{x}$, respectively. We use $\nabla_\theta f = \nabla_\theta f_\theta(x)$ to denote the derivative of parametrized model $f_\theta$ w.r.t. $\theta$ at the input $x$, and $\nabla_\theta \tilde{f}$ to denote the counterpart at the input $\tilde{x}$.

## 2 Background

For self-containedness, we first briefly review feedforward networks, residual networks and dual kernels associated with neural networks.

**Feedforward Networks**. We define an $L$-layer feedforward network (FFNet) $f(x)$ with ReLU activation in a recursive manner,

$$x_0 = x; \ x_\ell = \sqrt{\frac{2}{m}}\sigma_0(W_\ell x_{\ell-1}), \ \ell = 1, \cdots, L; \ f(x) = v^\top x_L, \tag{4}$$

where $W_1 \in \mathbb{R}^{m \times D}$ and $W_2, \cdots, W_L \in \mathbb{R}^{m \times m}$ are weight matrices, and $v \in \mathbb{R}^m$ is the output weight vector. For simplicity, we only consider feedforward networks with scalar outputs.

**Residual Networks**. We define an $L$-layer residual network (ResNet) $f(x)$ in a recursive manner,

$$x_0 = \sqrt{\frac{1}{m}}Ax; \ x_\ell = x_{\ell-1} + \alpha\sqrt{\frac{1}{m}}V_\ell\sigma_0\Big(\sqrt{\frac{2}{m}}W_\ell x_{\ell-1}\Big), \ \ell = 1, \cdots, L; \ f(x) = v^\top x_L, \tag{5}$$

where $W_\ell, V_\ell \in \mathbb{R}^{m \times m}$ for $\ell = 1, \cdots, L$, $A \in \mathbb{R}^{m \times D}$, $v \in \mathbb{R}^m$, and $\alpha = L^{-\gamma}$ is the scaling factor of the bottleneck layers. The scaling factor $\alpha$ is necessary for controlling the norm of $x_l$.

The network architecture in (5) is similar to the "pre-activation" shortcuts in [10], except that each bottleneck layer only contains one activation - between $W_\ell$ and $V_\ell$. We remove the activation of the input due to some technical issues (See more details in Section 3).

**Dual and Normalized Kernels**. The dual kernel technique was first proposed in [29] and motivated several follow-up works such as [28, 30]. Here we adopt the description in [28]. We use $K$ to denote a kernel function on the input space $\mathbb{R}^D$, i.e., $K : \mathbb{R}^D \times \mathbb{R}^D \to \mathbb{R}$. We denote

$$\Sigma(x, \tilde{x}) = \begin{pmatrix} K(x, x) & K(x, \tilde{x}) \\ K(\tilde{x}, x) & K(\tilde{x}, \tilde{x}) \end{pmatrix} \text{ and } N_\rho = \begin{pmatrix} 1 & \rho \\ \rho & 1 \end{pmatrix},$$

where $K \in \mathcal{F}$, $\rho \in \mathbb{R}$. Given an activation function $\phi : \mathbb{R} \to \mathbb{R}$, its dual activation function $\hat{\phi} : [-1, 1] \to [-1, 1]$ is defined to be $\hat{\phi}(\rho) = \mathbb{E}_{(X, \tilde{X}) \sim \mathcal{N}(0, N_\rho)}\phi(X)\phi(\tilde{X})$.

We then define the dual kernel as follows.

**Definition 1.** *We say that* $\Gamma_\phi(K) : \mathbb{R}^D \times \mathbb{R}^D \to \mathbb{R}$ *is the dual kernel of $K$ with respect to the activation $\phi$, if we have* $\Gamma_\phi(K)(x, \tilde{x}) = \mathbb{E}_{(X, \tilde{X}) \sim \mathcal{N}(0, \Sigma(x, \tilde{x}))}\phi(X)\phi(\tilde{X})$.

Note that $\Gamma_\phi(K)$ is also positive semi-definite. We also define the normalized kernel.

**Definition 2.** *We say that a kernel $K \in \mathcal{F}$ is normalized, if $K(x,x) = 1$ for all $x \in \mathbb{R}^D$. For a general kernel $K \in \mathcal{F}$, we define its normalized kernel by $\overline{K}$ where $\overline{K}(x,\tilde{x}) = \frac{K(x,\tilde{x})}{\sqrt{K(x,x)K(\tilde{x},\tilde{x})}}$.*

For *normalized ReLU function* $\sigma(z) = \sqrt{2}\max(0,z)$, [28] show $\hat{\sigma}(\rho) = \frac{\sqrt{1-\rho^2}+(\pi-\cos^{-1}(\rho))\rho}{\pi}$. Since $\sigma(z)$ is positive homogeneous, we have $\Gamma_\sigma(K)(x,\tilde{x}) = \sqrt{K(x,x)K(\tilde{x},\tilde{x})}\,\hat{\sigma}(\overline{K}(x,\tilde{x}))$. For *derivative of normalized ReLU function* $\sigma'(z) = \sqrt{2}\mathbb{I}_{\{z\geq 0\}}$, [28] show that $\widehat{\sigma'}(\rho) = \frac{\pi-\cos^{-1}(\rho)}{\pi}$. Since $\sigma'(z)$ is zeroth-order positive homogeneous, we have $\Gamma_{\sigma'}(K)(x,\tilde{x}) = \widehat{\sigma'}(\overline{K}(x,\tilde{x}))$. For more technical details of the dual kernel, we refer the readers to [28].

# 3 Neural Tangent Kernels of Deep Networks

There are two approaches to connecting neural networks to kernels: one is *Gaussian Process Kernel* (GP Kernel); the other is *Neural Tangent Kernel* (NTK). GP Kernel corresponds to the regime where the first $L$ layers are fixed after random initialization, and only the last layer is trained. Therefore, the first $L$ layers are essentially random feature mapping [31]. This is inconsistent with the practice, as the first $L$ layers should also be trained. In contrast, NTK corresponds to the regime where the first $L$ layers are also trained. For both GP Kernel and NTK, we consider the case when the width of the neural network goes to infinity. Due to space limit, we only provide some proof sketches for our theory, and all technical details are deferred to the appendix.

## 3.1 Feedforward Networks

We consider the Feedforward Network (FFNet) defined in (4), where $W_1 \in \mathbb{R}^{m \times D}$, $W_2, \cdots, W_L \in \mathbb{R}^{m \times m}$ and $v \in \mathbb{R}^m$ are all initialized as i.i.d. $\mathcal{N}(0,1)$ variables.[4] Given such random initialization, the outputs converge to a Gaussian process, as the width goes to infinity [32, 21]. Accordingly, the GP kernel is defined as follows.

**Proposition 1** ([28, 21]). *The GP kernel of the $L$-layer FFNet defined in (4) is*

$$K_0(x,\tilde{x}) = x^\top \tilde{x}; \ K_\ell(x,\tilde{x}) = \Gamma_\sigma(K_{\ell-1})(x,\tilde{x}), \ \ell = 1,\cdots,L. \tag{6}$$

**Theorem 1** ([28]). *For the FFNet defined in (4), there exists an absolute constant $C$, given the width $m \geq C\epsilon^{-2}L^2\log(8L/\delta)$, with probability at least $1-\delta$ over the randomness of the initialization, for input $x,\tilde{x}$ on the unit sphere, the inner product of the outputs of the $\ell$-th layer can be approximated by $K_\ell(x,\tilde{x})$, i.e.,*
$$|\langle x_\ell, \tilde{x}_\ell \rangle - K_\ell(x,\tilde{x})| \leq \epsilon, \ \text{for all } \ell = 1,\cdots,L.$$

The next proposition shows the NTK of this FFNet. Unlike the GP kernel, the NTK corresponds to the case when $\theta = (W_1, \cdots, W_L)$ are trained.

**Proposition 2** ([21]). *The NTK of the FFNet can be derived in terms of the GP kernels as*

$$\Omega_L(x,\tilde{x}) = \sum_{\ell=1}^{L}\left[K_{\ell-1}(x,\tilde{x})\prod_{i=\ell}^{L}\Gamma_{\sigma'}(K_{i-1})(x,\tilde{x})\right]. \tag{7}$$

Besides the asymptotic result, [22] further provide a nonasymptotic bound as follows.

**Theorem 2** ([22]). *For the FFNet defined in (4), when the width $m \geq CL^6\epsilon^{-4}\log(L/\delta)$, where $C$ is a constant, with probability at least $1-\delta$ over the initialization, for input $x,\tilde{x}$ on the unit sphere, the Neural Tangent Kernel can be approximated by $\Omega_L(x,\tilde{x})$, i.e.,*
$$\left|\langle \nabla_\theta f, \nabla_\theta \tilde{f} \rangle - \Omega_L(x,\tilde{x})\right| \leq L\epsilon.$$

[22] then showed that a sufficiently wide FFNet trained by gradient flow is close to the kernel regression predictor via its NTK.

**Remark 1.** *For self-containedness, we directly adopt the results from existing literature in this subsection. For more technical details on gradient flow and kernel ridge regression, we refer the readers to [28, 21, 22].*

## 3.2 Residual Networks

We consider the Residual Network (ResNet) in (5), where all parameters $(A, v, W_1, \cdots, W_L, V_1, \cdots, V_L)$ are independently initialized from the standard Gaussian

distribution. For simplicity, we only train $\theta = (W_1, \cdots, W_L, V_1, \cdots, V_L)$, but not $A$ or $v$, and the NTK of the ResNet is computed accordingly. Note that our theory can be naturally generalized to the setting where all parameters including $A$ and $v$ are trained, but the analysis will be more involved. Our next proposition derives the GP kernel of the ResNet.

**Proposition 3.** *The GP kernel of the ResNet is*
$$K_0(x, \tilde{x}) = x^\top \tilde{x}; \ K_\ell(x, \tilde{x}) = K_{\ell-1}(x, \tilde{x}) + \alpha^2 \Gamma_\sigma(K_{\ell-1})(x, \tilde{x}),$$
*where $\ell = 1, \cdots, L$, and $\alpha = L^{-\gamma}$ for $0.5 \le \gamma \le 1$.*

Proposition 3 demonstrates that each layer of the ResNet recursively "contributes" to the kernel in an incremental manner, which is quite different from that of the FFNet (shown in Proposition 1). Proposition 3 essentially provides a rigorous justification for the intuition discussed by [33]. Besides the above asymptotic result, we also derive a nonasymptotic bound as follows.

**Theorem 3.** *For the ResNet defined in (5), given two inputs on the unit sphere $x, \tilde{x} \in \mathbb{S}^{D-1}$, $\epsilon < 0.5$, and*
$$m \ge C\epsilon^{-2}L^{2-2\gamma}\log(36(L+1)/\delta),$$
*where $C$ is a constant and $0.5 \le \gamma \le 1$, with probability at least $1 - \delta$ over the randomness of the initialization, for all layers $\ell = 0, \cdots, L$ and $(x^{(1)}, x^{(2)}) \in \{(x, x), (x, \tilde{x}), (\tilde{x}, \tilde{x})\}$, we have*
$$|\langle x_\ell^{(1)}, x_\ell^{(2)} \rangle - K_\ell(x^{(1)}, x^{(2)})| \le \epsilon,$$
*where $K_\ell$ is recursively defined in Proposition 3.*

Theorem 3 implies that sufficiently wide residual networks are mimicking the GP kernel under proper conditions. The proof can be found in Appendix A. Next we present the NTK of the ResNet defined in (5) in the following proposition.

**Proposition 4.** *The NTK of the ResNet is $\Omega_L(x, \tilde{x}) = \alpha^2 \sum_{\ell=1}^L [B_{\ell+1}(x, \tilde{x})\Gamma_\sigma(K_{\ell-1})(x, \tilde{x}) + K_{\ell-1}(x, \tilde{x})B_{\ell+1}(x, \tilde{x})\Gamma_{\sigma'}(K_{\ell-1})(x, \tilde{x})]$, where $K_\ell$'s are defined in Proposition 3; $B_{L+1}(x, \tilde{x}) = 1$, and for $\ell = 1, \cdots, L$, $B_\ell$'s are defined as*
$$B_{\ell+1}(x, \tilde{x}) = B_{\ell+2}(x, \tilde{x}) + \alpha^2 B_{\ell+2}(x, \tilde{x})\Gamma_{\sigma'}(K_\ell)(x, \tilde{x}).$$

Proposition 4 implies that similar to what has been proved for the FFNet, the ResNet trained by gradient flow is also equivalent to the kernel regression predictor with some NTK. Note that Proposition 4 is an asymptotic result. We defer the proof, as it can be straightforwardly derived from the nonasymptotic bound as follows.

**Theorem 4.** *For the ResNet defined in (5), given two inputs on the unit sphere $x, \tilde{x} \in \mathbb{S}^{D-1}$, $\epsilon < 0.5$, and*
$$m \ge C\epsilon^{-4}L^{2-2\gamma}\big(\log(320(L^2+1)/\delta) + 1\big),$$
*where $C$ is a constant, with probability at least $1 - \delta$ over the randomness of the initialization, we have*
$$|\langle \nabla_\theta f, \nabla_\theta \tilde{f} \rangle - \Omega_L(x, \tilde{x})| \le 2L\alpha^2 \epsilon,$$
*where $\alpha = L^{-\gamma}$ with $\gamma \in [0.5, 1]$, $\Omega_L(x, \tilde{x})$ is defined in Proposition 4.*

*Proof Sketch of Proposition 4 and Theorem 4.* For simplicity, we use $\phi_W : \mathbb{R}^m \to \mathbb{R}^m$ to denote $\phi_W(z) = \sqrt{\frac{2}{m}}\sigma_0(Wz)$. Then its derivative w.r.t. $z$ is as follows, $\phi_W'(z) = \sqrt{\frac{2}{m}}D(Wz)W$, where $D(Wz)$ is an operator defined as $D(Wz) \equiv \mathrm{diag}(\sigma_0'(Wz)) = \mathrm{diag}([\mathbb{I}_{\{W_1, \cdot z \ge 0\}}, \cdots, \mathbb{I}_{\{W_m, \cdot z \ge 0\}}]^\top)$.

For simplicity, we denote $D_\ell = D(W_\ell x_{\ell-1})$, where $\ell = 1, 2, \cdots, L$. Note that $D_\ell$ is essentially the activation pattern of the $\ell$-th bottleneck layer on the input $x$. We denote $\tilde{D}_\ell$ for $\tilde{x}$ in a similar fashion. Then we have $\frac{\partial x_\ell}{\partial x_{\ell-1}} = I_m + \alpha\sqrt{\frac{1}{m}}V_\ell\sqrt{\frac{2}{m}}D_\ell W_\ell$. For $\ell = 1, \cdots, L$, we denote $b_{\ell+1} = \nabla_{x_\ell} f$. Then we have $b_{\ell+1} = \left(v^\top \frac{\partial x_L}{\partial x_{L-1}} \frac{\partial x_{L-1}}{\partial x_{L-2}} \cdots \frac{\partial x_{\ell+1}}{\partial x_\ell}\right)^\top$.

Combining all above derivations, we have $\nabla_{V_\ell} f = \frac{\alpha}{\sqrt{m}}b_{\ell+1} \cdot (\phi_{W_\ell}(x_{\ell-1}))^\top$, and $\nabla_{W_\ell} f = \frac{\alpha}{\sqrt{m}}\sqrt{\frac{2}{m}}D_\ell V_\ell^\top b_{\ell+1} \cdot x_{\ell-1}^\top$. Then we can derive the kernel $\sum_{\ell=1}^L \langle \nabla_{W_\ell} f, \nabla_{W_\ell} \tilde{f} \rangle + \sum_{\ell=1}^L \langle \nabla_{V_\ell} f, \nabla_{V_\ell} \tilde{f} \rangle$, where $\langle \nabla_{V_\ell} f, \nabla_{V_\ell} \tilde{f} \rangle = \alpha^2 \underbrace{\frac{1}{m}\langle b_{\ell+1}, \tilde{b}_{\ell+1} \rangle}_{T_{\ell,1}} \underbrace{\langle \phi_{W_\ell}(x_{\ell-1}), \phi_{W_\ell}(\tilde{x}_{\ell-1}) \rangle}_{T_{\ell,2}}$,

$$\langle \nabla_{W_\ell} f, \nabla_{W_\ell} \tilde{f} \rangle = \alpha^2 \underbrace{\langle x_{\ell-1}, \tilde{x}_{\ell-1} \rangle}_{T_{\ell,3}} \underbrace{\frac{2}{m^2} \tilde{b}_{\ell+1}^\top V_\ell \widetilde{D}_\ell D_\ell V_\ell^\top b_{\ell+1}}_{T_{\ell,4}} .$$ Note that the concentration of $T_{\ell,3}$ can

be shown by Theorem 3. We then show the concentration of $T_{\ell,1}$, $T_{\ell,2}$ and $T_{\ell,4}$, respectively.

For simplicity, we define two matrices for each layer,

$$\widehat{\Sigma}_\ell(x, \tilde{x}) = \begin{bmatrix} \langle x_\ell, x_\ell \rangle & \langle x_\ell, \tilde{x}_\ell \rangle \\ \langle \tilde{x}_\ell, x_\ell \rangle & \langle \tilde{x}_\ell, \tilde{x}_\ell \rangle \end{bmatrix}, \ \ \Sigma_\ell(x, \tilde{x}) = \begin{bmatrix} K_\ell(x, x) & K_\ell(x, \tilde{x}) \\ K_\ell(\tilde{x}, x) & K_\ell(\tilde{x}, \tilde{x}) \end{bmatrix}.$$

We define $\psi_\sigma : \mathcal{M}_+^2 \to \mathbb{R}$ as $\psi_\sigma(\Sigma) = \mathbb{E}_{(X, \tilde{X}) \sim \mathcal{N}(0, \Sigma)} \sigma(X) \sigma(\tilde{X})$ and $\psi_{\sigma'} : \mathcal{M}_+^2 \to \mathbb{R}$ as $\psi_{\sigma'}(\Sigma) = \mathbb{E}_{(X, \tilde{X}) \sim \mathcal{N}(0, \Sigma)} \sigma'(X) \sigma'(\tilde{X})$. Note $\Gamma_\sigma(K_{\ell-1}) = \psi_\sigma(\Sigma_{\ell-1})$ and $\Gamma_{\sigma'}(K_{\ell-1}) = \psi_{\sigma'}(\Sigma_{\ell-1})$.

The following lemmas are technical results and very involved. Please see Appendix B for details.

**Lemma 1.** *Suppose that for $\ell = 1, \cdots, L$,*

$$\|\widehat{\Sigma}_{\ell-1}(x, \tilde{x}) - \Sigma_{\ell-1}(x, \tilde{x})\|_{\max} \leq c\epsilon^2, \ \ m \geq C_1 \epsilon^{-2} L^{2-2\gamma} \big( \log(80L^2/\delta) + 1 \big), \quad (8)$$

*with probability at least $1 - 3\delta$, we have $|T_{\ell,1} - B_{\ell+1}(x, \tilde{x})| \leq c_1 \epsilon$, for $\ell = 1, \cdots, L$, where $C_1$, $c_1$, and $c$ are constants.*

**Lemma 2.** *Suppose (8) holds for $\ell = 1, \cdots, L$. With probability at least $1 - \delta$, we have $|T_{\ell,2} - \Gamma_\sigma(K_{\ell-1})(x, \tilde{x})| \leq c_2 \epsilon$, for $\ell = 1, \cdots, L$, where $C_2$ and $c_2$ are constants.*

**Lemma 3.** *Suppose that (8) holds for $\ell = 1, \cdots, L$. With probability at least $1 - 3\delta$, we have $|T_{\ell,4} - B_{\ell+1}(x, \tilde{x}) \Gamma_{\sigma'}(K_{\ell-1})(x, \tilde{x})| \leq c_3 \epsilon$, for $\ell = 1, \cdots, L$, where $c_3$ is a constant.*

We remark: (1) Lemma 1 is proved by reverse induction; (2) Lemma 2 exploits the concentration properties of $W_\ell$ and local Lipschitz properties of $\psi_\sigma$; (3) We prove Lemma 3 and Lemma 1 simultaneously with the Hölder continuity of $\psi_{\sigma'}$. Combining all results above, we complete Theorem 4. Moreover, taking $m \to \infty$, we have Proposition 4. □

## 4 Deep Feedforward v.s. Residual Networks

To compare the NTKs associated with deep FFNets and ResNets, we consider proper normalization, which avoids the kernel function blowing up or vanishing as the depth $L$ goes to infinity.

### 4.1 The Limiting NTK of the Feedforward Networks

Recall that the NTK of the $L$-layer FFNet defined in (4) is $\Omega_L(x, \tilde{x}) = \sum_{\ell=1}^L \big[ K_{\ell-1}(x, \tilde{x}) \cdot \prod_{i=\ell}^L \Gamma_{\sigma'}(K_{i-1})(x, \tilde{x}) \big]$. One can check that $\Omega_L(x, x) = L$ for all $x \in \mathbb{S}^{D-1}$. To avoid $\Omega_L(x, x) \to \infty$, as $L \to \infty$. We consider a normalized version as

$$\overline{\Omega}_L(x, \tilde{x}) = \frac{1}{L} \Omega_L(x, \tilde{x}).$$

We characterize the impact of the depth $L$ on the NTK in the following theorem.

**Theorem 5.** *For the NTK of the FFNet, as $L \to \infty$, given $x, \tilde{x} \in \mathbb{S}^{D-1}$ and $|1 - x^\top \tilde{x}| \geq \delta > 0$, where $\delta$ is a constant and does not scale with $L$, we have*

$$\left| \overline{\Omega}_L(x, \tilde{x}) - 1/4 \right| = \mathcal{O}\Big( \frac{\text{polylog}(L)}{L} \Big),$$

*When $x = \tilde{x}$, we have $\overline{\Omega}_L(x, \tilde{x}) = 1, \forall L$.*

*Proof Sketch of Theorem 5 .* The main challenge comes from the sophisticated recursion of the kernel. To handle the recursion, we employ the following bound.

**Lemma 4.** *When $L$ is large enough, we have*

$$\cos\left( \pi \left( 1 - \left( \frac{n}{n+1} \right)^{3 + \frac{\log(L)^2}{L}} \right) \right) \leq K_n(x, \tilde{x}) \leq \cos\left( \pi \left( 1 - \left( \frac{n + \log(L)^p}{n + \log(L)^p + 1} \right)^{3 - \frac{\log(L)^2}{L}} \right) \right),$$

*where $p$ is a positive constant depending on $\delta$.*

By Lemma 4, we can further bound $\prod_{i=\ell}^{L} \Gamma_{\sigma'}(K_{i-1}(x, \tilde{x}))$ by

$$\left(\frac{\ell - 1}{L}\right)^{3 + \frac{\log(L)^2}{L}} \leq \prod_{i=\ell}^{L} \Gamma_{\sigma'}(K_{i-1}(x, \tilde{x})) \leq \left(\frac{\ell + \log(L)^p - 1}{L + \log(L)^p}\right)^{3 - \frac{\log(L)^2}{L}} \tag{9}$$

Hence we can measure the rate of convergence. The detailed proof is the following. $\qquad\square$

As can be seen from Theorem 5, the NTK of the FFNet converges to a limiting form, i.e.,

$$\overline{\Omega}_\infty(x, \tilde{x}) = \lim_{L \to \infty} \overline{\Omega}_L(x, \tilde{x}) = \begin{cases} 1/4, & x \neq \tilde{x} \\ 1, & x = \tilde{x} \end{cases}.$$

For simplicity, we refer to $\overline{\Omega}_\infty$ as the limiting NTK of the FFNets.

The limiting NTK of the FFNets is actually a non-informative kernel. For example, we consider a kernel regression problem with $n$ independent observations $\{(x_i, y_i)\}_{i=1}^n$, where $x_i \in \mathbb{R}^D$ is the feature vector, and $y_i \in \mathbb{R}$ is the response. Without loss of generality, we assume that the training samples have been properly processed such that $x_i \neq x_j$ for $i \neq j$, and $\sum_{i=1}^n y_i = 0$. By the Representer theorem [34], we know that the kernel regression function can be represented by $f(\cdot) = \sum_{i=1}^n \beta_i \overline{\Omega}_\infty(x_i, \cdot)$. We then minimize the regularized empirical risk as follows.

$$\hat{\beta} = \min_\beta \|y - \widetilde{\Omega}\beta\|^2 + \lambda \beta^\top \widetilde{\Omega}\beta, \tag{10}$$

where $\beta = (\beta_1, ...\beta_n)^\top \in \mathbb{R}^n$, $y = (y_1, ..., y_n)^\top \in \mathbb{R}^n$, $\widetilde{\Omega} \in \mathbb{R}^{n \times n}$ with $\widetilde{\Omega}_{ij} = \overline{\Omega}_\infty(x_i, x_j)$, and $\lambda$ is the regularization parameter and usually very small for large $n$. One can check that (10) admits a closed form solution $\hat{\beta} = (\widetilde{\Omega} + \lambda I_n)^{-1} y$. Note that we have $\widetilde{\Omega} + \lambda I_n = 1/4 J_n + (\lambda + 3/4) I_n$, which is the sum of a diagonal matrix and a rank-one matrix and $J_n$ is $n \times n$ all-ones matrix. By Sherman – Morrison formula

$$(A + uv^\top)^{-1} = A^{-1} - \frac{A^{-1}uv^\top A^{-1}}{1 + v^\top A^{-1}u}, \text{ we have } \hat{\beta} = \frac{1}{\lambda + 3/4}\left(I_n - \frac{1}{n + 4\lambda + 3}J_n\right)y.$$

Then we further have $f(x_j) = \sum_{i=1}^n \hat{\beta}_i \overline{\Omega}_\infty(x_i, x_j) = \frac{3}{4\lambda+3} y_j$.

As can be seen, for sufficiently large $n$ and sufficiently small $\lambda$, we have $f(x_j) \approx y_j$, which means that we can fit the training data well. However, for an unseen data point $x^*$, where $x^* \neq x_1, ..., x_n$, the regression function $f$ always gives an output 0, i.e.,

$$f(x^*) = \sum_{i=1}^n \hat{\beta}_i \overline{\Omega}_\infty(x_i, x^*) = \frac{1}{4}\sum_{i=1}^n \hat{\beta}_i = 0.$$

This indicates that the function class induced by the limiting NTK of the FFNets $\overline{\Omega}_\infty$ is not learnable.

### 4.2 The Limiting NTK of the Residual Networks

Recall that the infinite-width NTK of the $L$-layer ResNet is

$$\Omega_L(x, \tilde{x}) = \alpha^2 \sum_{\ell=1}^{L} \left[B_{\ell+1}(x, \tilde{x})\Gamma_\sigma(K_{\ell-1})(x, \tilde{x}) + K_{\ell-1}(x, \tilde{x})B_{\ell+1}(x, \tilde{x})\Gamma_{\sigma'}(K_{\ell-1})(x, \tilde{x})\right],$$

where $B_{L+1}(x, \tilde{x}) = 1$ and for $\ell = 1, .., L - 1$, $B_{\ell+1}(x, \tilde{x}) = \prod_{i=\ell}^{L-1}(1 + \alpha^2 \Gamma_{\sigma'}(K_i)(x, \tilde{x}))$. One can check that for $x \in \mathbb{S}^{D-1}$, $\Omega_L(x, x) = 2L\alpha^2(1 + \alpha^2)^{L-1}$.

Different from the NTK of the FFNet, $\Omega_L(x, x) \to 0$ as $L \to \infty$. Therefore, we also consider the normalized NTK for the ResNet to prevent the kernel from vanishing. Specifically, the normalized NTK of the ResNet on $\mathbb{S}^{D-1} \times \mathbb{S}^{D-1}$, $\overline{\Omega}_L(x, \tilde{x})$, is defined as follows,

$$\frac{1/(2L)}{(1 + \alpha^2)^{L-1}} \sum_{\ell=1}^{L} \left[B_{\ell+1}(x, \tilde{x})\Gamma_\sigma(K_{\ell-1})(x, \tilde{x}) + K_{\ell-1}(x, \tilde{x})B_{\ell+1}(x, \tilde{x})\Gamma_{\sigma'}(K_{\ell-1})(x, \tilde{x})\right]. \tag{11}$$

We then analyze the limiting NTK of the ResNets. Recall that $\alpha = L^{-\gamma}$. Our next theorem only considers $\gamma = 1$, i.e., $\alpha = 1/L$.

**Theorem 6.** *For the NTK of the ResNet, as $L \to \infty$, given $\alpha = \frac{1}{L}$ and $x, \tilde{x} \in \mathbb{S}^{D-1}$ such that $|1 - x^\top\tilde{x}| \geq \delta > 0$, where $\delta$ is a constant and does not scale with L, we have*

$$\left|\overline{\Omega}_L(x, \tilde{x}) - \overline{\Omega}_1(x, \tilde{x})\right| = \mathcal{O}\left(1/L\right),$$

*where $\overline{\Omega}_1(x, \tilde{x}) = \frac{1}{2}\left(\hat{\sigma}(x^\top\tilde{x}) + x^\top\tilde{x} \cdot \hat{\sigma}'(x^\top\tilde{x})\right)$.*

*Proof Sketch of Theorem 6.* The main technical challenge here is also handling the recursion. Specifically, we denote $K_{\ell,L}$ to be the $\ell$-th layer of the GP kernel when the depth is $L$, which is originally denoted by $K_\ell(x,\tilde{x})$. Let $S_0 = K_0(x,\tilde{x})$ and $S_{\ell,L} = \frac{K_{\ell,L}}{(1+\alpha^2)^\ell} = \frac{K_{\ell,L}}{(1+1/L^2)^\ell}$. We have $\Gamma_\sigma(K_{\ell,L}) = (1+\alpha^2)^\ell \hat{\sigma}(S_{\ell,L})$ and $\Gamma_{\sigma'}(K_{\ell,L}) = \widehat{\sigma'}(S_{\ell,L})$. We rewrite the recursion of $K_{\ell,L}$ as $S_{\ell,L} = \frac{S_{\ell-1,L} + \alpha^2 \hat{\sigma}(S_{\ell-1,L})}{(1+\alpha^2)} \geq S_{\ell-1,L}$, which eases the technical difficulty. However, the proof is still highly involved, and more details can be found in Appendix E. □

Note that we do not consider $\gamma = 0.5$ for technical concerns, as $\overline{\Omega}_L(x,\tilde{x})$ in (11) becomes very complicated to compute, as $L \to \infty$. Also we find that considering $\gamma = 1$ is sufficient to provide us new theoretical insights on ResNets (See more details in Section 5).

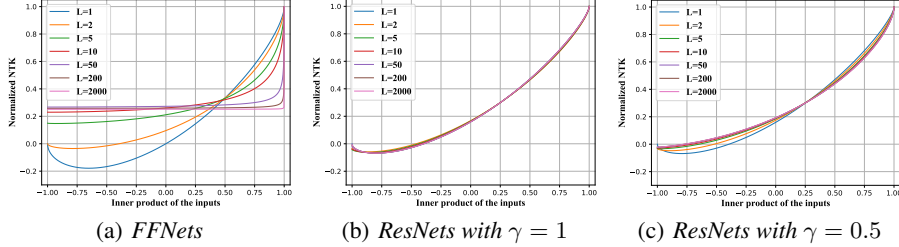

|  (a) *FFNets* | (b) *ResNets with $\gamma = 1$* | (c) *ResNets with $\gamma = 0.5$* |

Figure 1: *Normalized Neural Tangent Kernels Associated with Different Deep Networks.*

Different from FFNets, the class of functions induced by the NTKs of the ResNets does not significantly change, as the depth $L$ increases. Surprisingly, we actually have $\overline{\Omega}_\infty = \overline{\Omega}_1$ for $\alpha = 1/L$, i.e., infinitely deep and 1-layer ResNets induce the same NTK. To further visualize such a difference, we plot the NTKs of the ResNets in Fig. 1(b) and 1(c) for $\alpha = 1/L$ and $\alpha = 1/\sqrt{L}$, respectively. As can be seen, the increase of the depth yields very small changes to the NTKs. This partially explains why increasing the depth of the ResNet does not significantly deteriorate the generalization.

Moreover, as long as $x \neq \tilde{x}$, i.e., $\langle x,\tilde{x}\rangle \neq 1$, the limiting NTK of the FFNets always yields $1/4$ regardless how different $x$ is from $\tilde{x}$. In contrast, the residual networks do not suffer from this drawback. The limiting NTK of the ResNets can greatly distinguish the difference between $x$ and $\tilde{x}$, e.g., $\langle x,\tilde{x}\rangle = -0.5$, 0, and 0.5 yield different values. Therefore, for an unseen data point, the corresponding regression model does not always output 0, which is in sharp contrast to that of the limiting NTK of the FFNets.

## 5 Experiments

We demonstrate the generalization properties of the kernel regression based on the NTKs of the FFNets and the ResNets with varying depths. Our experiments follow similar settings to [22, 23]. We adopt two widely used data sets – MNIST [35] and CIFAR10 [36], which are popular in existing literature. Note that both MNIST and CIFAR10 contains 10 classes of images. For simplicity, we select 2 classes out of 10 (digits "0" and "8" for MNIST, categories "airplane" and "ship" for CIFAR10), respectively, which results in two binary classification problems, denoted by MNIST2 and CIFAR2.

Similar to [22, 23], we use the kernel regression model for classification. Specifically, given the training data $(x_1,y_1),\cdots,(x_n,y_n)$, where $x_i \in \mathbb{R}^D$ and $y_i \in \{-1,+1\}$ for $i = 1,...,n$, we compute the kernel matrix $\tilde{K} = [\tilde{K}_{ij}]_{i,j=1}^n$ using the NTKs associated with the FFNets and the ResNets, where $\tilde{K}_{ij} = \overline{\Omega}_L(x_i,x_j)$. Then we compute the kernel regression function $f(x) = \sum_{i=1}^n \alpha_i \overline{\Omega}_L(x,x_i)$, where $[\alpha_1,...,\alpha_n]^\top = (\tilde{K} + \lambda I)^{-1}y$, $y = [y_1,...,y_n]^\top$ and $\lambda = 0.1/n$ is a very small constant. We predict the label of $x$ to be $\text{sign}(f(x))$.

Our experiments adopt the NTKs associated with three network architectures: (1) FFNets, (2) ResNets ($\gamma = 0.5$) and (3) ResNets ($\gamma = 1$). We set $n = 200$ and $n = 2000$. For each data set, we randomly select $n$ training data points ($n/2$ for each class) and 2000 testing data points (1000 for each class). When training the kernel regression models, we normalize all training data points to have zero mean and unit norm. We repeat the procedure for 20 simulations. We find that the training errors of all simulations ($L$ varies from 1 to 2000) are 0.0, which means that all NTK-based models are sufficient to overfit the training data, regardless $n = 200$ or $n = 2000$. The test accuracies of the kernel regression models with different kernels and depths are shown in Figure 2.

As can be seen, the test accuracies of the kernel regression models of ResNets (both $\gamma = 0.5$ and $\gamma = 1$) are not sensitive to the depth. In contrast, the test accuracies of the kernel regression models of the FFNets significantly decrease, as the depth $L$ increases. Especially when the sample size is small ($n = 200$), the kernel regression models behave like random guess for both MNIST2 and CIFAR2 when $L \geq 1000$. This is consistent with our analysis.

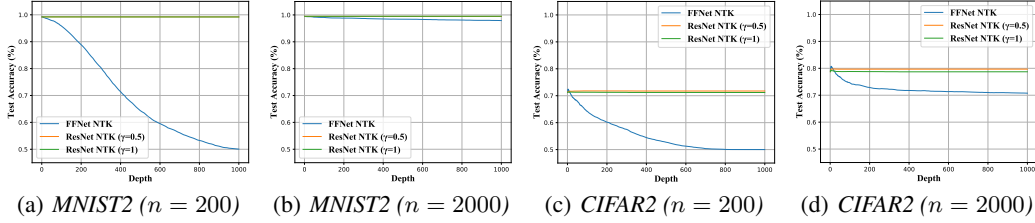

(a) *MNIST2 (n = 200)*    (b) *MNIST2 (n = 2000)*    (c) *CIFAR2 (n = 200)*    (d) *CIFAR2 (n = 2000)*

Figure 2: *Test accuracies of the kernel regression models evaluated on MNIST2 and CIFAR2.*

Next we provide numerical verifications for our theorems. For Theorem 4, we randomly initialize the ResNet with width=500, scaling factor $\gamma = 1$ and depth $L = 5, 10, 100, 300$, and then calculate the inner product of the Jacobians of the ResNet for two different inputs as in the definition of NTK. We repeat the procedure for 500 times and plot the mean value (black cross) and the 1/4, 3/4 quantiles ("I"-shape line) of the sampled random NTKs and the theoretical NTK value in Fig. 3(a), which shows the two results match very well. For Theorem 5 and Theorem 6, Fig. 3(b) and Fig. 3(c) show that $\lim_{L \to \infty} |\overline{\Omega}_L(x, \tilde{x}) - 1/4| \cdot L/\log(L) \approx$ constant and $\lim_{L \to \infty} |\overline{\Omega}_L(x, \tilde{x}) - \overline{\Omega}_1(x, \tilde{x})| \cdot L \approx$ constant with $x^\top \tilde{x} = K_0$ chosen at 9 points.

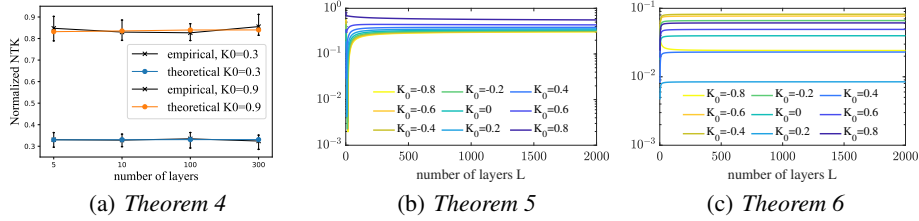

(a) *Theorem 4*    (b) *Theorem 5*    (c) *Theorem 6*

Figure 3: *Verification of main theorems. (a) Theorem 4, $m = 500$ and scaling $\gamma = 1$; (b) Theorem 5, y-axis is $|\overline{\Omega}_L(x, \tilde{x}) - 1/4| \cdot L/\log(L)$; (c) Theorem 6, y-axis is $|\overline{\Omega}_L(x, \tilde{x}) - \overline{\Omega}_1(x, \tilde{x})| \cdot L$*

# 6    Discussion

We discuss the NTK of the ResNet in more details. We remark unless specified, the NTK mentioned below indicates the normalized NTK.

Our theory shows the function class induced by the NTK of the deep ResNet asymptotically converges to that by the NTK of the 1-layer ResNet, as the depth increases. This indicates that the complexity of such a function class is not significantly different from that by the NTK of the 1-layer ResNet, for large enough $L$. Thus, the generalization gap does not significantly increase, as $L$ increases.

On the other hand, our experiments suggest that, as illustrated in Figure 4, the NTK of the ResNet with $\gamma = 1$ actually achieves the best testing accuracy for CIFAR2 when $L = 2$. The accuracy slightly decreases as $L$ increases, and becomes stable when $L \geq 9$. For the NTK of the ResNet with $\gamma = 0.5$, the accuracy achieves the best when $L \approx 15$, and becomes stable for $L \geq 15$. Such evidence suggests that the function class induced by the NTKs of the ResNets with large $L$ and large $\gamma$ are possibly not as flexible as those by the NTKs of the deep ResNets with small $L$ and small $\gamma$.

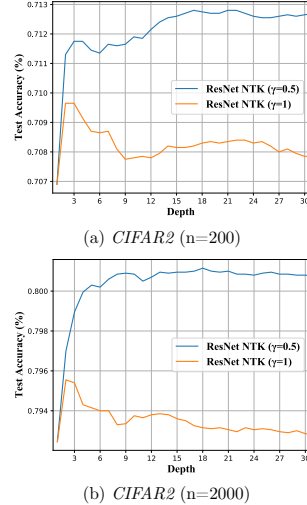

(a) *CIFAR2* (n=200)

(b) *CIFAR2* (n=2000)

Figure 4: *Test accuracies of the kernel regression models evaluated on CIFAR2.*

Existing literature connects overparameterized neural networks to NTKs only under some very specific regime. Practical neural networks, however, are trained under more complicated regimes. Therefore, there still exists a significant theoretical gap between NTKs and practical neural networks. For example, Theorem 6 shows that the NTK of the infinitely deep ResNet is identical to that of the 1-layer ResNet, while practical ResNets often show better generalization performance, as the depth increases. Also, we do not consider batch norm in our networks but refer to [37] if necessary. We will leave these challenges for future investigation.

## Broader Impact

This paper makes a significant contribution to extending the frontier of deep learning theory, and increases the intellectual rigor. To the best of our knowledge, our results are the first one for analyzing the effect of depth on the generalization of neural tangent kernels (NTKs). Moreover, our results are also the first one establishing the non-asymptotic bounds for NTKs of ResNets when all but the last layers are trained, which enables us to successfully analyze the generalization properties of ResNets through the perspective of NTK. This is in sharp contrast to the existing impractical theoretical results for NTKs of ResNets, which either only apply to an over-simplified structure of ResNets or only deal with the case when the last layer is trained.

## Acknowledgement

Molei Tao was partially supported by NSF DMS-1847802 and ECCS-1936776 and Yuqing Wang was partially supported by NSF DMS-1847802.

## Footnotes

[2]More precisely, our analysis considers the regime, where the widths go to infinity first, and then the depths go to infinity. See more details in Section 4.

[3]Although the ReLU function $\sigma_0$ is not differentiable at 0, we call $\sigma_0'$ derivative for notational convenience.

[4]In general, the weight matrices do not need to be square matrices, nor do they need to be of the same size.

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
