[Supplementary Material]

# A  Proof of GP Kernels of ResNets

## A.1  Notation and Main Idea

For a fixed pair of inputs $x$ and $\tilde{x}$, we introduce two matrices for each layer

$$\hat{\Sigma}_\ell(x, \tilde{x}) = \begin{bmatrix} \langle x_\ell, x_\ell \rangle & \langle x_\ell, \tilde{x}_\ell \rangle \\ \langle \tilde{x}_\ell, x_\ell \rangle & \langle \tilde{x}_\ell, \tilde{x}_\ell \rangle \end{bmatrix},$$

and

$$\Sigma_\ell(x, \tilde{x}) = \begin{bmatrix} K_\ell(x, x) & K_\ell(x, \tilde{x}) \\ K_\ell(\tilde{x}, x) & K_\ell(\tilde{x}, \tilde{x}) \end{bmatrix}.$$

$\hat{\Sigma}_\ell(x, \tilde{x})$ is the empirical Gram matrix of the outputs of the $\ell$-th layer, while $\Sigma_\ell(x, \tilde{x})$ is the infinite-width version. Theorem 3 says that with high probability, for each layer $\ell$, the difference of these two matrices measured by the entry-wise $L_\infty$ norm (denoted by $\| \cdot \|_{\max}$) is small.

The idea is to bound how much the $\ell$-th layer magnifies the input error to the output. Specifically, if the outputs of $(\ell - 1)$-th layer satisfy

$$\left\| \hat{\Sigma}_{\ell-1}(x, \tilde{x}) - \Sigma_{\ell-1}(x, \tilde{x}) \right\|_{\max} \leq \tau,$$

we hope to prove that with high probability over the randomness of $W_\ell$ and $V_\ell$, we have

$$\left\| \hat{\Sigma}_\ell(x, \tilde{x}) - \Sigma_\ell(x, \tilde{x}) \right\|_{\max} \leq \left( 1 + \mathcal{O}\left(\frac{1}{L}\right) \right) \tau.$$

Then the theorem is proved by first showing that w.h.p. $\left\| \hat{\Sigma}_0(x, \tilde{x}) - \Sigma_0(x, \tilde{x}) \right\|_{\max} \leq (1 + \mathcal{O}(1/L))^{-L} \epsilon$ and then applying the result above for each layer.

## A.2  Lemmas

We introduce the following lemmas. The first lemma shows the boundedness of $K_\ell(x, \tilde{x})$.

**Lemma 5.** *For the ResNet defined in Eqn. (5), $K_\ell(x, x) = (1 + \alpha^2)^\ell$ for all $x \in \mathbb{S}^{D-1}$, $\ell = 0, 1, \cdots, L$. Also $K_\ell(x, x)$ is bounded uniformly when $0.5 \leq \gamma \leq 1$.*

Recall that $\phi_{W_\ell}(z) = \sqrt{\frac{2}{m}} \sigma_0(W_\ell z)$. Since $W_\ell$ is Gaussian, we know that $\phi_{W_\ell}(x_{\ell-1})$ and $\phi_{W_\ell}(\tilde{x}_{\ell-1})$ are both sub-Gaussian random vectors over the randomness of $W_\ell$. Then their inner product enjoys sub-exponential property.

**Lemma 6** (Sub-exponential concentration)**.** *With probability at least $1 - \delta'$ over the randomness of $W_\ell \sim \mathcal{N}(0, I)$, when $m \geq c' \log(6/\delta')$, the following hold simultaneously*

$$\left| \langle \phi_{W_\ell}(x_{\ell-1}), \phi_{W_\ell}(\tilde{x}_{\ell-1}) \rangle - \psi_\sigma(\hat{\Sigma}_{\ell-1}(x, \tilde{x})) \right| \leq \sqrt{\frac{c' \log(6/\delta')}{m}} \|x_{\ell-1}\| \|\tilde{x}_{\ell-1}\|, \qquad (12)$$

$$\left| \|\phi_{W_\ell}(x_{\ell-1})\|^2 - \|x_{\ell-1}\|^2 \right| \leq \sqrt{\frac{c' \log(6/\delta')}{m}} \|x_{\ell-1}\|^2, \qquad (13)$$

$$\left| \|\phi_{W_\ell}(\tilde{x}_{\ell-1})\|^2 - \|\tilde{x}_{\ell-1}\|^2 \right| \leq \sqrt{\frac{c' \log(6/\delta')}{m}} \|\tilde{x}_{\ell-1}\|^2. \qquad (14)$$

**Lemma 7** (Locally Lipschitzness, based on [28])**.** *$\psi_\sigma$ is $(1 + \frac{1}{\pi}(\frac{r}{\mu})^2)$-Lipschitz w.r.t. $\max$ norm in $\mathcal{M}_{\mu, r} = \left\{ \begin{bmatrix} a & b \\ b & c \end{bmatrix} | a, c \in [\mu - r, \mu + r]; ac - b^2 > 0 \right\}$ for all $\mu > 0$, $0 < r \leq \mu/2$. That means, if (i). $\|\hat{\Sigma}_{\ell-1}(x, \tilde{x}) - \Sigma_{\ell-1}(x, \tilde{x})\|_{\max} \leq \tau$ and (ii). $K_{\ell-1}(x, x) = K_{\ell-1}(\tilde{x}, \tilde{x}) = \mu$, for $\tau \leq \mu/2$, we have*

$$\left| \psi_\sigma(\hat{\Sigma}_{\ell-1}(x, \tilde{x})) - \psi_\sigma(\Sigma_{\ell-1}(x, \tilde{x})) \right| \leq \left( 1 + \frac{1}{\pi}\left(\frac{\tau}{\mu}\right)^2 \right) \tau.$$

## A.3  Proof of Theorem 3

*Proof.* In this proof, we also show the following hold with the same probability.

1. For $\ell = 0, 1, \cdots, L$, $\|x_\ell\|$ and $\|\tilde{x}_\ell\|$ are bounded by an absolute constant $C_1$ ($C_1 = 4$).

2. For $\ell = 1, \cdots, L$, $\|\phi_{W_\ell}(x_{\ell-1})\|$ and $\|\phi_{W_\ell}(\tilde{x}_{\ell-1})\|$ are bounded by an absolute constant $C_2$ ($C_2 = 8$).

3. $\left|\langle\phi_{W_\ell}(x_{\ell-1}^{(1)}), \phi_{W_\ell}(x_{\ell-1}^{(2)})\rangle - \Gamma_\sigma(K_{\ell-1})(x^{(1)}, x^{(2)})\right| \leq 2\epsilon$ for all $\ell = 1, \cdots, L$ and $(x^{(1)}, x^{(2)}) \in \{(x, x), (x, \tilde{x}), (\tilde{x}, \tilde{x})\}$.

We focus on the $\ell$-th layer. Let $\tau = \left\|\hat{\Sigma}_{\ell-1}(x, \tilde{x}) - \Sigma_{\ell-1}(x, \tilde{x})\right\|_{\max}$. Recall that $\Gamma_\sigma(K_{\ell-1})(x, \tilde{x}) = \psi_\sigma(\Sigma_{\ell-1}(x, \tilde{x})) = \mathbb{E}_{(X, \tilde{X}) \sim \mathcal{N}(0, \Sigma_{\ell-1}(x, \tilde{x}))} \sigma(X)\sigma(\tilde{X})$. Then

$$K_\ell(x, \tilde{x}) = K_{\ell-1}(x, \tilde{x}) + \alpha^2 \psi_\sigma(\Sigma_{\ell-1}(x, \tilde{x})).$$

Since $x_\ell = x_{\ell-1} + \frac{\alpha}{\sqrt{m}} V_\ell \phi_{W_\ell}(x_{\ell-1})$, we have

$$\langle x_\ell, \tilde{x}_\ell\rangle = \langle x_{\ell-1}, \tilde{x}_{\ell-1}\rangle + \frac{\alpha^2}{m}\langle V_\ell\phi_{W_\ell}(x_{\ell-1}), V_\ell\phi_{W_\ell}(\tilde{x}_{\ell-1})\rangle$$
$$+ \alpha\frac{1}{\sqrt{m}}\left(\langle V_\ell\phi_{W_\ell}(x_{\ell-1}), \tilde{x}_{\ell-1}\rangle + \langle V_\ell\phi_{W_\ell}(\tilde{x}_{\ell-1}), x_{\ell-1}\rangle\right)$$
$$= \langle x_{\ell-1}, \tilde{x}_{\ell-1}\rangle + \alpha^2 P + \alpha(Q + R),$$

where

$$P \equiv \frac{1}{m}\langle V_\ell\phi_{W_\ell}(x_{\ell-1}), V_\ell\phi_{W_\ell}(\tilde{x}_{\ell-1})\rangle,$$
$$Q \equiv \frac{1}{\sqrt{m}}\left(\langle V_\ell\phi_{W_\ell}(x_{\ell-1}), \tilde{x}_{\ell-1}\rangle\right),$$
$$R \equiv \frac{1}{\sqrt{m}}\left(\langle V_\ell\phi_{W_\ell}(\tilde{x}_{\ell-1}), x_{\ell-1}\rangle\right).$$

Under the randomness of $V_\ell$, $P$ is sub-exponential, and $Q$ and $R$ are Gaussian random variables. Therefore, for a given $\delta_0$, if $m \geq c_0 \log(2/\delta_0)$, with probability at least $1 - \delta_0$ over the randomness of $V_\ell$, we have

$$\left|P - \langle\phi_{W_\ell}(x_{\ell-1}), \phi_{W_\ell}(\tilde{x}_{\ell-1})\rangle\right| \leq \|\phi_{W_\ell}(x_{\ell-1})\|\|\phi_{W_\ell}(\tilde{x}_{\ell-1})\|\sqrt{\frac{c_0\log(2/\delta_0)}{m}}; \qquad (15)$$

for a given $\tilde{\delta}$, with probability at least $1 - 2\tilde{\delta}$ over the randomness of $V_\ell$, we have

$$|Q| \leq \|\phi_{W_\ell}(x_{\ell-1})\|\|\tilde{x}_{\ell-1}\|\sqrt{\frac{\tilde{c}\log(2/\tilde{\delta})}{m}}, \qquad (16)$$

and

$$|R| \leq \|\phi_{W_\ell}(\tilde{x}_{\ell-1})\|\|x_{\ell-1}\|\sqrt{\frac{\tilde{c}\log(2/\tilde{\delta})}{m}}, \qquad (17)$$

where $c_0, \tilde{c} > 0$ are absolute constants.

Using the above result and Lemma 6 and setting $\delta_0 = \tilde{\delta} = \frac{\delta}{18(L+1)}$, $\delta' = \frac{\delta}{6(L+1)}$, when $m \geq C\log(36(L+1)/\delta)$, we have (15), (16), (17), (12), (13), and (14) hold with probability at least $1 - \frac{\delta}{3(L+1)}$.

Recall that $\tau = \left\|\hat{\Sigma}_{\ell-1}(x, \tilde{x}) - \Sigma_{\ell-1}(x, \tilde{x})\right\|_{\max}$. Conditioned on $\tau < 0.5$, we have

$$\|x_{\ell-1}\|^2 \leq K_{\ell-1}(x, x) + \tau \leq (1 + \alpha^2)^L + \tau \leq e + \tau.$$

Similarly we can show $\|\tilde{x}_{\ell-1}\|^2$ is bounded by $e + \tau$. By (13) and (14) we have $\|\phi_{W_\ell}(x_{\ell-1})\|^2 \leq 2\|x_{\ell-1}\|^2$ and $\|\phi_{W_\ell}(\tilde{x}_{\ell-1})\|^2 \leq 2\|\tilde{x}_{\ell-1}\|^2$, which are both bounded.

Then
$$\left|\langle x_\ell, \tilde{x}_\ell \rangle - \left(\alpha^2 \psi_\sigma(\Sigma_{\ell-1}(x,\tilde{x})) + K_{\ell-1}(x,\tilde{x})\right)\right|$$

$$\leq \tau + \alpha^2 \left(P - \psi_\sigma(\Sigma_{\ell-1}(x,\tilde{x}))\right) + \alpha(|Q| + |R|)$$

$$\leq \tau + \alpha^2 \left|P - \langle \phi_{W_\ell}(x_{\ell-1}), \phi_{W_\ell}(\tilde{x}_{\ell-1})\rangle\right| + \alpha\sqrt{\frac{\tilde{c}\log(2/\tilde{\delta})}{m}}\left(\|\phi_{W_\ell}(\tilde{x}_{\ell-1})\|\|x_{\ell-1}\| + \|\phi_{W_\ell}(x_{\ell-1})\|\|\tilde{x}_{\ell-1}\|\right)$$

$$+ \alpha^2\left|\psi_\sigma(\hat{\Sigma}_{\ell-1}(x,\tilde{x})) - \psi_\sigma(\Sigma_{\ell-1}(x,\tilde{x}))\right| + \alpha^2\left|\langle \phi_{W_\ell}(x_{\ell-1}), \phi_{W_\ell}(\tilde{x}_{\ell-1})\rangle - \psi_\sigma(\hat{\Sigma}_{\ell-1}(x,\tilde{x}))\right|$$

$$\leq \tau + (\alpha^2 + \alpha)\sqrt{\frac{C_3\log(36(L+1)/\delta)}{m}} + \alpha^2\tau\left(1 + \frac{1}{\pi}\left(\frac{\tau}{K_{\ell-1}(x,x)}\right)^2\right)$$

$$\leq \tau + (\alpha^2 + \alpha)\sqrt{\frac{C_3\log(36(L+1)/\delta)}{m}} + \alpha^2\tau\left(1 + \frac{1}{4\pi}\right).$$

When $\alpha = \frac{1}{L^\gamma}$, $\gamma \in [0.5, 1]$, we have $\alpha^2 \leq 1/L$. Then when
$$m \geq \frac{C_3 L^{2(1-\gamma)}\log(36(L+1)/\delta)}{\tau^2},$$
we have
$$\left|\langle x_\ell, \tilde{x}_\ell\rangle - K_\ell(x,\tilde{x})\right| \leq \tau + \frac{4}{L}\tau.$$
As a byproduct, we have
$$\left|\langle \phi_{W_\ell}(x_{\ell-1}), \phi_{W_\ell}(\tilde{x}_{\ell-1})\rangle - \psi_\sigma(\Sigma_{\ell-1}(x,\tilde{x}))\right|$$

$$\leq \sqrt{\frac{C_4\log(36(L+1)/\delta)}{m}} + \left(1 + \frac{1}{\pi}\left(\frac{\tau}{\mu}\right)^2\right)\tau \leq 2\tau.$$
Repeat the above for $(x_{\ell-1}, x_{\ell-1})$ and $(\tilde{x}_{\ell-1}, \tilde{x}_{\ell-1})$, we have with probability at least $1 - \delta/(L+1)$ over the randomness of $V_\ell$ and $W_\ell$,
$$\left\|\hat{\Sigma}_{\ell-1}(x,\tilde{x}) - \Sigma_{\ell-1}(x,\tilde{x})\right\|_{\max} \leq \tau \Rightarrow$$
$$\left\|\hat{\Sigma}_\ell(x,\tilde{x}) - \Sigma_\ell(x,\tilde{x})\right\|_{\max} \leq (1 + 4/L)\tau. \tag{18}$$
Finally, when $m \geq \frac{C_5\log(6(L+1)/\delta)}{(\epsilon/e^4)^2}$, with probability at least $1 - \delta/(L+1)$ over the randomness of $A$, we have
$$\left\|\hat{\Sigma}_0(x,\tilde{x}) - \Sigma_0(x,\tilde{x})\right\|_{\max} \leq \epsilon/e^4.$$
Then the result follows by successively using (18). $\qquad\square$

## A.4 proof of lemma 7

*Proof.* [28] showed that
$$\left\|\nabla\psi_\sigma\begin{bmatrix} a & b \\ b & c \end{bmatrix}\right\|_1 = \frac{1}{2}\frac{a+c}{\sqrt{ac}}\left|\hat{\sigma}\left(\frac{b}{\sqrt{ac}}\right) - \frac{b}{\sqrt{ac}}\hat{\sigma}'\left(\frac{b}{\sqrt{ac}}\right)\right| + \hat{\sigma}'\left(\frac{b}{\sqrt{ac}}\right).$$
When $a, c \in [\mu - r, \mu + r]$, we have
$$\frac{1}{2}\frac{a+c}{\sqrt{ac}} = \frac{1}{2}\left(\sqrt{\frac{a}{c}} + \sqrt{\frac{c}{a}}\right) \leq \frac{1}{2}\left(\sqrt{\frac{\mu+r}{\mu-r}} + \sqrt{\frac{\mu-r}{\mu+r}}\right) = \left(1 - \left(\frac{r}{\mu}\right)^2\right)^{-1/2} \leq 1 + \left(\frac{r}{\mu}\right)^2.$$
The last inequality holds when $r < \frac{\mu}{2}$.

Define $\rho = \frac{b}{\sqrt{ac}}$, we have $\rho \in [-1, 1]$. Then
$$\|\nabla\phi_\sigma\|_1 \leq \left(1 + \left(\frac{r}{\mu}\right)^2\right)\left|\hat{\sigma}(\rho) - \rho\hat{\sigma}'(\rho)\right| + \hat{\sigma}'(\rho)$$

$$= \left(1 + \left(\frac{r}{\mu}\right)^2\right)\left|\frac{\sqrt{1-\rho^2}}{\pi}\right| + 1 - \frac{\cos^{-1}\rho}{\pi}$$

$$\leq \frac{\sqrt{1-\rho^2}}{\pi} + 1 - \frac{\cos^{-1}\rho}{\pi} + \frac{1}{\pi}\left(\frac{r}{\mu}\right)^2$$

$$\leq 1 + \frac{1}{\pi}\left(\frac{r}{\mu}\right)^2.$$

$\qquad\square$

# B Proof of Theorem 4

## B.1 Notation and Main Idea

We already know that when the network width $m$ is large enough, $\langle x_{\ell-1}, \tilde{x}_{\ell-1} \rangle \approx K_{\ell-1}(x, \tilde{x})$, and $\langle \phi_{W_\ell}(x_{\ell-1}), \phi_{W_\ell}(\tilde{x}_{\ell-1}) \rangle \approx \Gamma_\sigma(K_{\ell-1})(x, \tilde{x})$.

Next we need to show the concentration of the inner product of $\frac{b_\ell}{\sqrt{m}}$ and $\frac{\tilde{b}_\ell}{\sqrt{m}}$. We define two matrices for each layer

$$\hat{\Theta}_\ell(x, \tilde{x}) = \frac{1}{m} \begin{bmatrix} \langle b_\ell, b_\ell \rangle & \langle b_\ell, \tilde{b}_\ell \rangle \\ \langle \tilde{b}_\ell, b_\ell \rangle & \langle \tilde{b}_\ell, \tilde{b}_\ell \rangle \end{bmatrix},$$

and

$$\Theta_\ell(x, \tilde{x}) = \begin{bmatrix} B_\ell(x, x) & B_\ell(x, \tilde{x}) \\ B_\ell(\tilde{x}, x) & B_\ell(\tilde{x}, \tilde{x}) \end{bmatrix}.$$

Recall that

$$b_\ell = \alpha \sqrt{\frac{1}{m}} \sqrt{\frac{2}{m}} W_\ell^\top D_\ell V_\ell^\top b_{\ell+1} + b_{\ell+1}.$$

We aim to show that when $\|\hat{\Theta}_{\ell+1}(x, \tilde{x}) - \Theta_{\ell+1}(x, \tilde{x})\|_{\max} \leq \tau$, with high probability over the randomness of $W_\ell$ and $V_\ell$, we have $\|\hat{\Theta}_\ell(x, \tilde{x}) - \Theta_\ell(x, \tilde{x})\|_{\max} \leq (1 + \mathcal{O}(1/L))\tau$. Notice that $b_{\ell+1}$ and $\tilde{b}_{\ell+1}$ contain the information of $W_\ell$ and $V_\ell$; they are not independent. Nevertheless we can decompose the randomness of $W_\ell$ and $V_\ell$ to show the concentration. This technique is also used in [22].

## B.2 Lemmas

In this part we introduce some useful lemmas. The first one shows the property of the step activation function.

**Lemma 8** (Property of $\sigma'$). *[22]*

*(1). Sub-Gaussian concentration. With probability at least $1 - \delta$ over the randomness of $W_\ell$, we have*

$$\left| \frac{2}{m} \operatorname{Tr}(D_\ell \tilde{D}_\ell) - \psi_{\sigma'}(\hat{\Sigma}_{\ell-1}(x, \tilde{x})) \right| \leq \sqrt{\frac{c \log(2/\delta)}{m}}.$$

*(2). Holder continuity. Fix $\mu > 0, 0 < r \leq \mu$. For all $A, B \in \mathcal{M}_{\mu,r} = \left\{ \begin{bmatrix} a & b \\ b & c \end{bmatrix} \middle| a, c \in [\mu - r, \mu + r]; ac - b^2 > 0 \right\}$, if $\|A - B\|_{\max} \leq (\mu - r)\epsilon^2$, then*

$$|\psi_{\sigma'}(A) - \psi_{\sigma'}(B)| \leq \epsilon.$$

The following lemma shows that regardless the fact that $b_{\ell+1}$ and $\tilde{b}_{\ell+1}$ depend on $V_\ell$, we can treat $V_\ell$ as a Gaussian matrix independent of $b_{\ell+1}$ and $\tilde{b}_{\ell+1}$ when the network width is large enough.

**Lemma 9.** *Assume the following inequality hold simultaneously for all $\ell = 1, 2, \cdots, L$*

$$\left\| \frac{1}{\sqrt{m}} W_\ell \right\| \leq C, \quad \left\| \frac{1}{\sqrt{m}} V_\ell \right\| \leq C.$$

*Fix an $\ell$. Further assume that*

$$\|\hat{\Theta}_{\ell+1}(x, \tilde{x}) - \Theta_{\ell+1}(x, \tilde{x})\|_{\max} \leq 1.$$

*When $m \geq \max\{\frac{C}{\epsilon^2}(1 + \log \frac{6}{\delta}), \frac{C}{\epsilon^2} \log \frac{8L}{\delta'}, cL^{2-2\gamma} \log \frac{8L}{\delta'}\}$, the following holds for all $(x^{(1)}, x^{(2)}) \in \{(x, x), (x, \tilde{x}), (\tilde{x}, \tilde{x})\}$ with probability at least $1 - \delta - \delta'$*

$$\left| \frac{2}{m} \frac{b_{\ell+1}^{(1)}}{\sqrt{m}}^\top V_\ell D_\ell^{(1)} D_\ell^{(2)} V_\ell^\top \frac{b_{\ell+1}^{(2)}}{\sqrt{m}} - \langle \frac{b_{\ell+1}^{(1)}}{\sqrt{m}}, \frac{b_{\ell+1}^{(2)}}{\sqrt{m}} \rangle \frac{2}{m} \operatorname{Tr}(D_\ell^{(1)} D_\ell^{(2)}) \right| \leq \epsilon.$$

The following lemma shows the same thing for $W_\ell$ as $V_\ell$ in Lemma 9.

**Lemma 10.** *Assume the conditions and the results of Lemma 9 hold.*

*(1). When $m \geq \max\{\frac{C}{\epsilon^2}(1 + \log \frac{6}{\delta}), \frac{C}{\epsilon^2} \log \frac{8L}{\delta'}, cL^{2-2\gamma} \log \frac{8L}{\delta'}\}$, the following holds for all $(x^{(1)}, x^{(2)}) \in \{(x, x), (x, \tilde{x}), (\tilde{x}, \tilde{x})\}$ with probability at least $1 - \delta - \delta'$*

$$\left| \frac{1}{m} \frac{2}{m} \langle W_\ell^\top D_\ell^{(1)} V_\ell^\top \frac{b_{\ell+1}^{(1)}}{\sqrt{m}}, W_\ell^\top D_\ell^{(2)} V_\ell^\top \frac{b_{\ell+1}^{(2)}}{\sqrt{m}} \rangle - \frac{2}{m} \langle D_\ell^{(1)} V_\ell^\top \frac{b_{\ell+1}^{(1)}}{\sqrt{m}}, D_\ell^{(2)} V_\ell^\top \frac{b_{\ell+1}^{(2)}}{\sqrt{m}} \rangle \right| \leq \epsilon.$$

*(2).* *When* $m \geq \max\{\frac{C}{\epsilon^2}\log\frac{16L}{\tilde{\delta}}, cL^{2-2\gamma}\log\frac{16L}{\tilde{\delta}}\}$, *for all* $(x^{(1)}, x^{(2)}) \in \{(x,x),(x,\tilde{x}),(\tilde{x},x),(\tilde{x},\tilde{x})\}$, *the following holds with probability at least* $1 - \tilde{\delta}$

$$\left|\frac{1}{m}\sqrt{\frac{1}{m}}\sqrt{\frac{2}{m}}\langle W_\ell^\top D_\ell^{(1)} V_\ell^\top b_{\ell+1}^{(1)}, b_{\ell+1}^{(2)}\rangle\right| \leq \tilde{\epsilon}.$$

## B.3  Proof of Theorem 4

*Proof.* In this proof we are going to prove that when $m$ satisfies the assumption, with probability at least $1 - \delta_0$, the following hold for $\ell = 1, \cdots, L$.

$$\left|\frac{1}{\alpha^2}\langle\nabla_{V_\ell}f, \nabla_{V_\ell}\tilde{f}\rangle - B_{\ell+1}(x,\tilde{x})\Gamma_\sigma(K_{\ell-1})(x,\tilde{x})\right| \leq \epsilon_0,$$

$$\left|\frac{1}{\alpha^2}\langle\nabla_{W_\ell}f, \nabla_{W_\ell}\tilde{f}\rangle - K_{\ell-1}(x,\tilde{x})B_{\ell+1}(x,\tilde{x})\Gamma_{\sigma'}(K_{\ell-1})(x,\tilde{x})\right| \leq \epsilon_0.$$

We break the proof into several steps. Each step is based on the result of the previous steps. Note that the absolute constants $c$ and $C$ may vary throughout the proof.

### Step 1. Norm Control of the Gaussian Matrices

With probability at least $1 - \delta_1$, when $m > c\log\frac{4L}{\delta_1}$, one can show that the following hold simultaneously for all $\ell = 1, 2, \cdots, L$ [38]

$$\left\|\frac{1}{\sqrt{m}}W_\ell\right\| \leq C, \quad \left\|\frac{1}{\sqrt{m}}V_\ell\right\| \leq C.$$

### Step 2. Concentration of the GP kernels

By Theorem 3, with probability at least $1 - \delta_2$, when

$$m \geq \frac{C}{\epsilon_2^4}L^{2-2\gamma}\log\frac{36(L+1)}{\delta_2},$$

we have

1. For $\ell = 0, \cdots, L$, $\left\|\Sigma_\ell(x,\tilde{x}) - \hat{\Sigma}_\ell(x,\tilde{x})\right\|_{\max} \leq c\epsilon_2^2$;

2. For $\ell = 0, 1, \cdots, L$, $\|x_\ell\|$ and $\|\tilde{x}_\ell\|$ are bounded by an absolute constant $C_1$ ($C_1 = 4$);

3. For $\ell = 1, \cdots, L$, $\|\phi_{W_\ell}(x_{\ell-1})\|$ and $\|\phi_{W_\ell}(\tilde{x}_{\ell-1})\|$ are bounded by an absolute constant $C_2$ ($C_2 = 8$);

4. $\left|\langle\phi_{W_\ell}(x_{\ell-1}^{(1)}), \phi_{W_\ell}(x_{\ell-1}^{(2)})\rangle - \Gamma_\sigma(K_{\ell-1})(x^{(1)}, x^{(2)})\right| \leq 2c\epsilon_2^2$ for all $\ell = 1, \cdots, L$ and $(x^{(1)}, x^{(2)}) \in \{(x,x),(x,\tilde{x}),(\tilde{x},\tilde{x})\}$.

### Step 3. Concentration of $\sigma'$

By Lemma 8, when $m \geq \frac{C}{\epsilon_2^2}\log\frac{6L}{\delta_3}$, with probability at least $1 - \delta_3$, for all $\ell = 1, 2, \cdots, L$ and $(x^{(1)}, x^{(2)}) \in \{(x,x),(x,\tilde{x}),(\tilde{x},\tilde{x})\}$, we have

$$\left|\frac{2}{m}\mathrm{Tr}(D_\ell^{(1)}D_\ell^{(2)}) - \Gamma_{\sigma'}(K_{\ell-1})(x^{(1)}, x^{(2)})\right| \leq \sqrt{\frac{c\log(6L/\delta_3)}{m}} + \sqrt{2\left\|\hat{\Sigma}_{\ell-1}(x,\tilde{x}) - \Sigma_{\ell-1}(x,\tilde{x})\right\|_{\max}} \leq \epsilon_2.$$

### Step 4. Concentration of $B_\ell$

Recall that

$$b_{\ell+1} = \left(v^\top \frac{\partial x_L}{\partial x_{L-1}}\frac{\partial x_{L-1}}{\partial x_{L-2}}\cdots\frac{\partial x_{\ell+1}}{\partial x_\ell}\right)^\top.$$

We have

$$b_{L+1} = v,$$

and for $\ell = 1, 2, \cdots, L-1$,

$$b_{\ell+1} = \frac{\partial x_{\ell+1}}{\partial x_\ell}^\top b_{\ell+2} = \alpha\sqrt{\frac{1}{m}}\sqrt{\frac{2}{m}}W_{\ell+1}^\top D_{\ell+1}V_{\ell+1}^\top b_{\ell+2} + b_{\ell+2}.$$

Following the same idea in Thm 3, we prove by induction. First of all, for $b_{L+1}$, we have
$$\Theta_{L+1}(x,\tilde{x}) = \begin{bmatrix} 1 & 1 \\ 1 & 1 \end{bmatrix}, \hat{\Theta}_{L+1}(x,\tilde{x}) = \frac{\|v\|^2}{m}\begin{bmatrix} 1 & 1 \\ 1 & 1 \end{bmatrix}.$$ Then by Bernstein inequality [39], with probability at least $1 - \frac{\delta_4}{L}$, when $m \geq \frac{C}{\epsilon_4^2}\log\frac{2L}{\delta_4}$, we have
$$\left| \frac{\|v\|^2}{m} - 1 \right| \leq \epsilon_4.$$

Fix $\ell \in \{2, 3, \cdots, L\}$. Assume that
$$\left\| \hat{\Theta}_{\ell+1}(x,\tilde{x}) - \Theta_{\ell+1}(x,\tilde{x}) \right\|_{\max} \leq \tau \leq 1,$$
we hope to prove with high probability,
$$\left\| \hat{\Theta}_{\ell}(x,\tilde{x}) - \Theta_{\ell}(x,\tilde{x}) \right\|_{\max} \leq (1 + \mathcal{O}(1/L))\tau.$$

First write
$$\frac{1}{m}\langle b_\ell^{(1)}, b_\ell^{(2)} \rangle = \frac{1}{m}\langle b_{\ell+1}^{(1)}, b_{\ell+1}^{(2)} \rangle + \alpha^2 P + \alpha(Q + R),$$
where
$$P = \frac{1}{m}\frac{2}{m}\langle W_\ell^\top D_\ell^{(1)} V_\ell^\top \frac{b_{\ell+1}^{(1)}}{\sqrt{m}}, W_\ell^\top D_\ell^{(2)} V_\ell^\top \frac{b_{\ell+1}^{(2)}}{\sqrt{m}} \rangle,$$
$$Q = \frac{1}{m}\sqrt{\frac{1}{m}}\sqrt{\frac{2}{m}}\langle W_\ell^\top D_\ell^{(1)} V_\ell^\top b_{\ell+1}^{(1)}, b_{\ell+1}^{(2)} \rangle,$$
$$R = \frac{1}{m}\sqrt{\frac{1}{m}}\sqrt{\frac{2}{m}}\langle W_\ell^\top D_\ell^{(2)} V_\ell^\top b_{\ell+1}^{(2)}, b_{\ell+1}^{(1)} \rangle.$$

Then
$$\left| \frac{1}{m}\langle b_\ell^{(1)}, b_\ell^{(2)} \rangle - (B_{\ell+1}(x^{(1)}, x^{(2)}) + \alpha^2 B_{\ell+1}(x^{(1)}, x^{(2)})\Gamma_{\sigma'}(K_{\ell-1})(x^{(1)}, x^{(2)})) \right|$$
$$\leq \left| \frac{1}{m}\langle b_{\ell+1}^{(1)}, b_{\ell+1}^{(2)} \rangle - B_{\ell+1}(x^{(1)}, x^{(2)}) \right| + \alpha^2 \left| P - B_{\ell+1}(x^{(1)}, x^{(2)})\Gamma_{\sigma'}(K_{\ell-1})(x^{(1)}, x^{(2)}) \right| + \alpha|Q| + \alpha|R|$$
$$\leq \tau + \alpha^2 \left| P - \frac{2}{m}\langle D_\ell^{(1)} V_\ell^\top \frac{b_{\ell+1}^{(1)}}{\sqrt{m}}, D_\ell^{(2)} V_\ell^\top \frac{b_{\ell+1}^{(2)}}{\sqrt{m}} \rangle \right|$$
$$+ \alpha^2 \left| \frac{2}{m}\langle D_\ell^{(1)} V_\ell^\top \frac{b_{\ell+1}^{(1)}}{\sqrt{m}}, D_\ell^{(2)} V_\ell^\top \frac{b_{\ell+1}^{(2)}}{\sqrt{m}} \rangle - \langle \frac{b_{\ell+1}^{(1)}}{\sqrt{m}}, \frac{b_{\ell+1}^{(2)}}{\sqrt{m}} \rangle \frac{2}{m}\mathrm{Tr}(D_\ell^{(1)} D_\ell^{(2)}) \right|$$
$$+ \alpha^2 \left| \langle \frac{b_{\ell+1}^{(1)}}{\sqrt{m}}, \frac{b_{\ell+1}^{(2)}}{\sqrt{m}} \rangle - B_{\ell+1}(x^{(1)}, x^{(2)}) \right| \left| \frac{2}{m}\mathrm{Tr}(D_\ell^{(1)} D_\ell^{(2)}) \right|$$
$$+ \alpha^2 \left| B_{\ell+1}(x^{(1)}, x^{(2)}) \right| \left| \frac{2}{m}\mathrm{Tr}(D_\ell^{(1)} D_\ell^{(2)}) - \Gamma_{\sigma'}(K_{\ell-1})(x^{(1)}, x^{(2)}) \right|$$
$$+ \alpha|Q| + \alpha|R|.$$

In Lemma 9 and Lemma 10, set $\tilde{\epsilon} = cL^{\gamma-1}\tau$, $\epsilon = c\tau$, $\delta = \tilde{\delta} = \delta' = \delta_4/5L$. When $m \geq \max\{\frac{C}{\tau^2}(1 + \log\frac{30L}{\delta_4}), \frac{C}{\tau^2}\log\frac{40L^2}{\delta_4}, \frac{C}{\tau^2}L^{2-2\gamma}\log\frac{80L^2}{\delta_4}, cL^{2-2\gamma}\log\frac{80L^2}{\delta_4}\}$, with probability at least $1 - \frac{\delta_4}{L}$, the results of Lemma 9 and Lemma 10 hold. Then for all $(x^{(1)}, x^{(2)}) \in \{(x,x), (x,\tilde{x}), (\tilde{x},\tilde{x})\}$,
$$\left| \frac{1}{m}\langle b_\ell^{(1)}, b_\ell^{(2)} \rangle - B_\ell(x^{(1)}, x^{(2)}) \right| \leq \tau + \alpha^2 c\tau + \alpha^2 c\tau + \alpha^2 2\tau + \alpha^2 e\epsilon_2 + 2\alpha c L^{a-1}\tau$$
$$\leq \tau(1 + \mathcal{O}(1/L)). \quad (\text{Set } \epsilon_2 \leq c\tau.)$$

By taking union bound, with probability at least $1 - \delta_4$, we have for all $\ell = 1, 2, \cdots, L$,
$$\|\hat{\Theta}_{\ell+1}(x,\tilde{x}) - \Theta_{\ell+1}(x,\tilde{x})\|_{\max} \leq (1 + \mathcal{O}(1/L))^L \epsilon_4 \leq C\epsilon_4.$$
Meanwhile, we have for all $(x^{(1)}, x^{(2)}) \in \{(x,x), (x,\tilde{x}), (\tilde{x},\tilde{x})\}$ and $\ell = 1, \cdots, L$,
$$\left| \frac{2}{m}\langle D_\ell^{(1)} V_\ell^\top \frac{b_{\ell+1}^{(1)}}{\sqrt{m}}, D_\ell^{(2)} V_\ell^\top \frac{b_{\ell+1}^{(2)}}{\sqrt{m}} \rangle - B_{\ell+1}(x^{(1)}, x^{(2)})\Gamma_{\sigma'}(K_{\ell-1})(x^{(1)}, x^{(2)}) \right| \leq (2+c)\tau + e\epsilon_2 \leq C\epsilon_4.$$

**Step 5. Summary**

Using previous results, for all $\ell$, we have

$$\left| \frac{1}{\alpha^2} \langle \nabla_{V_\ell} f, \nabla_{V_\ell} \tilde{f} \rangle - B_{\ell+1} \Gamma_\sigma(K_{\ell-1}) \right|$$

$$\leq \left| \frac{1}{m} \langle b_{\ell+1}, \tilde{b}_{\ell+1} \rangle - B_{\ell+1} \right| \cdot |\langle \phi_{W_\ell}(x_{\ell-1}), \phi_{W_\ell}(\tilde{x}_{\ell-1}) \rangle| + |B_{\ell+1}| \cdot |\langle \phi_{W_\ell}(x_{\ell-1}), \phi_{W_\ell}(\tilde{x}_{\ell-1}) \rangle - \Gamma_\sigma(K_{\ell-1})|$$

$$\leq C\epsilon_4 + C\epsilon_2^2,$$

and

$$\left| \frac{1}{\alpha^2} \langle \nabla_{W_\ell} f, \nabla_{W_\ell} \tilde{f} \rangle - K_{\ell-1} B_{\ell+1} \Gamma_{\sigma'}(K_{\ell-1}) \right|$$

$$\leq \left| \frac{1}{m} \langle x_{\ell-1}, \tilde{x}_{\ell-1} \rangle - K_{\ell-1} \right| \cdot \left| \frac{2}{m} \tilde{b}_{\ell+1}^\top V_\ell \widetilde{D}_\ell D_\ell V_\ell^\top b_{\ell+1} \right| + |K_{\ell-1}| \cdot \left| \frac{2}{m} \tilde{b}_{\ell+1}^\top V_\ell \widetilde{D}_\ell D_\ell V_\ell^\top b_{\ell+1} - B_{\ell+1} \Gamma_{\sigma'}(K_{\ell-1}) \right|$$

$$\leq C\epsilon_2^2 + C\epsilon_4.$$

To sum up, by choosing $\epsilon_4 = c\epsilon_0$, $\epsilon_2 = c\epsilon_4$, and $\delta_1 = \delta_2 = \delta_3 = \delta_4 = \delta_0/4$, then with probability at least $1 - \delta_0$, when

$$m \geq \frac{C}{\epsilon_0^4} L^{2-2\gamma} \left( \log \frac{320(L^2+1)}{\delta_0} + 1 \right)$$

$$\geq \max \left\{ c \log \frac{16L}{\delta_0}, \frac{C}{\epsilon_0^4} L^{2-2\gamma} \log \frac{144(L+1)}{\delta_0}, \frac{C}{\epsilon_0^2} \log \frac{24L}{\delta_0}, \right.$$

$$\left. \frac{C}{\epsilon_0^2} \log \frac{8L}{\delta_0}, \frac{C}{\epsilon_0^2} (1 + \log \frac{120L}{\delta_0}), \frac{C}{\epsilon_0^2} \log \frac{160L^2}{\delta_0}, \frac{C}{\epsilon_0^2} L^{2-2\gamma} \log \frac{320L^2}{\delta_0}, cL^{2-2\gamma} \log \frac{320L^2}{\delta_0^4} \right\},$$

the desired results hold.

$\square$

## C  Proofs of the Lemmas

### C.1  Supporting lemmas

**Lemma 11.** *Define $G = [\phi_{W_\ell}(x_{\ell-1}), \phi_{W_\ell}(\tilde{x}_{\ell-1})]$, and $\Pi_G^\perp$ as the orthogonal projection onto the orthogonal complement of the column space of $G$. when $m \geq 1 + \log \frac{6}{\delta}$, the following holds with probability at least $1 - \delta$ for all $(x^{(1)}, x^{(2)}) \in \{(x, x), (x, \tilde{x}), (\tilde{x}, \tilde{x})\}$,*

$$\left| \frac{2}{m} \frac{b_{\ell+1}^{(1)}}{\sqrt{m}}^\top V_\ell \Pi_G^\perp D_\ell^{(1)} D_\ell^{(2)} \Pi_G^\perp V_\ell^\top \frac{b_{\ell+1}^{(2)}}{\sqrt{m}} - \langle \frac{b_{\ell+1}^{(1)}}{\sqrt{m}}, \frac{b_{\ell+1}^{(1)}}{\sqrt{m}} \rangle \frac{2}{m} \operatorname{Tr}(D_\ell^{(1)} D_\ell^{(2)}) \right| \leq (4 + 4\sqrt{2}) M \sqrt{\frac{1 + \log \frac{6}{\delta}}{m}},$$

*where*

$$M = \max \left\{ \frac{\|b_{\ell+1}\|^2}{m}, \frac{\|\tilde{b}_{\ell+1}\|^2}{m} \right\}.$$

*proof of Lemma 11.* We prove the lemma on any realization of $(A, W_1, V_1, \cdots, W_{\ell-1}, V_{\ell-1}, W_\ell, W_{\ell+1}, V_{\ell+1}, \cdots, W_L, V_L, v)$, $V_\ell \phi_{W_\ell}(x_{\ell-1})$ and $V_\ell \phi_{W_\ell}(\tilde{x}_{\ell-1})$, and consider the remaining randomness of $V_\ell$. In this case, $D_\ell, \widetilde{D}_\ell, b_{\ell+1}$ and $\tilde{b}_{\ell+1}$ are fixed.

One can show that conditioned on the realization of $V_\ell G$ (whose "degree of freedom" is $2m$), $V_\ell \Pi_G^\perp$ is identically distributed as $\widetilde{V}_\ell \Pi_G^\perp$, where $\widetilde{V}_\ell$ is an i.i.d. copy of $V_\ell$. The remaining $m^2 - 2m$ "degree of freedom" is enough for a good concentration. For the proof of this result, we refer the readers to Lemma E.3 in [22].

Denote $T = \Pi_G^\perp D_\ell^{(1)} D_\ell^{(2)} \Pi_G^\perp$,

$$S = \begin{bmatrix} \widetilde{V}_\ell^\top \frac{b_{\ell+1}^{(1)}}{\sqrt{m}} \\ \widetilde{V}_\ell^\top \frac{b_{\ell+1}^{(2)}}{\sqrt{m}} \end{bmatrix}.$$

We know that $S$ is a $2m$-dimensional Gaussian random vector, and

$$S \sim \mathcal{N} \left( 0, \begin{bmatrix} \langle \frac{b_{\ell+1}^{(1)}}{\sqrt{m}}, \frac{b_{\ell+1}^{(1)}}{\sqrt{m}} \rangle I_m & \langle \frac{b_{\ell+1}^{(1)}}{\sqrt{m}}, \frac{b_{\ell+1}^{(2)}}{\sqrt{m}} \rangle I_m \\ \langle \frac{b_{\ell+1}^{(2)}}{\sqrt{m}}, \frac{b_{\ell+1}^{(1)}}{\sqrt{m}} \rangle I_m & \langle \frac{b_{\ell+1}^{(2)}}{\sqrt{m}}, \frac{b_{\ell+1}^{(2)}}{\sqrt{m}} \rangle I_m \end{bmatrix} \right).$$

Then there exists a matrix $P \in \mathbb{R}^{2m \times 2m}$ such that
$$PP^\top = \begin{bmatrix} \langle \frac{b_{\ell+1}^{(1)}}{\sqrt{m}}, \frac{b_{\ell+1}^{(1)}}{\sqrt{m}} \rangle I_m & \langle \frac{b_{\ell+1}^{(1)}}{\sqrt{m}}, \frac{b_{\ell+1}^{(2)}}{\sqrt{m}} \rangle I_m \\ \langle \frac{b_{\ell+1}^{(2)}}{\sqrt{m}}, \frac{b_{\ell+1}^{(1)}}{\sqrt{m}} \rangle I_m & \langle \frac{b_{\ell+1}^{(2)}}{\sqrt{m}}, \frac{b_{\ell+1}^{(2)}}{\sqrt{m}} \rangle I_m \end{bmatrix},$$
and $S \stackrel{d}{=} P\xi, \xi \sim \mathcal{N}(0, I_{2m})$.

Thus
$$\frac{b_{\ell+1}^{(1)}}{\sqrt{m}}^\top \widetilde{V}_\ell \Pi_G^\perp D_\ell^{(1)} D_\ell^{(2)} \Pi_G^\perp \widetilde{V}_\ell^\top \frac{b_{\ell+1}^{(2)}}{\sqrt{m}} \stackrel{d}{=} \xi^\top P^\top \begin{bmatrix} I_m \\ 0 \end{bmatrix}^\top T \begin{bmatrix} 0 \\ I_m \end{bmatrix} P\xi = \frac{1}{2}\xi^\top P^\top \begin{bmatrix} 0 & T \\ T & 0 \end{bmatrix} P\xi.$$

We have
$$\left\| \frac{1}{2} P^\top \begin{bmatrix} 0 & T \\ T & 0 \end{bmatrix} P \right\| \leq \frac{1}{2} \|P^\top\| \cdot \|P\| \cdot \left\| \begin{bmatrix} 0 & T \\ T & 0 \end{bmatrix} \right\|$$
$$= \frac{1}{2} \|PP^\top\| \cdot \|T\|$$
$$\leq \frac{1}{2} \left\| \begin{bmatrix} \langle \frac{b_{\ell+1}^{(1)}}{\sqrt{m}}, \frac{b_{\ell+1}^{(1)}}{\sqrt{m}} \rangle I_m & \langle \frac{b_{\ell+1}^{(1)}}{\sqrt{m}}, \frac{b_{\ell+1}^{(2)}}{\sqrt{m}} \rangle I_m \\ \langle \frac{b_{\ell+1}^{(2)}}{\sqrt{m}}, \frac{b_{\ell+1}^{(1)}}{\sqrt{m}} \rangle I_m & \langle \frac{b_{\ell+1}^{(2)}}{\sqrt{m}}, \frac{b_{\ell+1}^{(2)}}{\sqrt{m}} \rangle I_m \end{bmatrix} \right\| \left\| \Pi_G^\perp \right\| \left\| D_\ell^{(1)} \right\| \left\| D_\ell^{(2)} \right\| \left\| \Pi_G^\perp \right\|$$
$$\leq \frac{\langle \frac{b_{\ell+1}^{(1)}}{\sqrt{m}}, \frac{b_{\ell+1}^{(1)}}{\sqrt{m}} \rangle + \langle \frac{b_{\ell+1}^{(2)}}{\sqrt{m}}, \frac{b_{\ell+1}^{(2)}}{\sqrt{m}} \rangle}{2} \leq M.$$
And $\left\| \frac{1}{2} P^\top \begin{bmatrix} 0 & T \\ T & 0 \end{bmatrix} P \right\|_F \leq \sqrt{2m} M.$

Then by the Hanson-Wright Inequality for Gaussian chaos [40], we have with probability at least $1 - \delta/3$,
$$\frac{2}{m} \left| \frac{b_{\ell+1}^{(1)}}{\sqrt{m}}^\top \widetilde{V}_\ell \Pi_G^\perp D_\ell^{(1)} D_\ell^{(2)} \Pi_G^\perp \widetilde{V}_\ell^\top \frac{b_{\ell+1}^{(2)}}{\sqrt{m}} - \mathbb{E}_{\widetilde{V}_\ell} \left[ \frac{b_{\ell+1}^{(1)}}{\sqrt{m}}^\top \widetilde{V}_\ell \Pi_G^\perp D_\ell^{(1)} D_\ell^{(2)} \Pi_G^\perp \widetilde{V}_\ell^\top \frac{b_{\ell+1}^{(2)}}{\sqrt{m}} \right] \right|$$
$$\leq \frac{4}{m} \left( \sqrt{2m} M \sqrt{\log \frac{6}{\delta}} + M \log \frac{6}{\delta} \right),$$
Furthermore, we have
$$\mathbb{E}_{\widetilde{V}_\ell} \left[ \frac{b_{\ell+1}^{(1)}}{\sqrt{m}}^\top \widetilde{V}_\ell \Pi_G^\perp D_\ell^{(1)} D_\ell^{(2)} \Pi_G^\perp \widetilde{V}_\ell^\top \frac{b_{\ell+1}^{(2)}}{\sqrt{m}} \right] = \langle \frac{b_{\ell+1}^{(1)}}{\sqrt{m}}, \frac{b_{\ell+1}^{(1)}}{\sqrt{m}} \rangle \operatorname{Tr}(\Pi_G^\perp D_\ell^{(1)} D_\ell^{(2)}).$$
Thus
$$\left| \frac{2}{m} \mathbb{E}_{\widetilde{V}_\ell} \left[ \frac{b_{\ell+1}^{(1)}}{\sqrt{m}}^\top \widetilde{V}_\ell \Pi_G^\perp D_\ell^{(1)} D_\ell^{(2)} \Pi_G^\perp \widetilde{V}_\ell^\top \frac{b_{\ell+1}^{(2)}}{\sqrt{m}} \right] - \langle \frac{b_{\ell+1}^{(1)}}{\sqrt{m}}, \frac{b_{\ell+1}^{(1)}}{\sqrt{m}} \rangle \frac{2}{m} \operatorname{Tr}(D_\ell^{(1)} D_\ell^{(2)}) \right|$$
$$= \frac{2}{m} \left| \langle \frac{b_{\ell+1}^{(1)}}{\sqrt{m}}, \frac{b_{\ell+1}^{(1)}}{\sqrt{m}} \rangle \operatorname{Tr}(\Pi_G D_\ell^{(1)} D_\ell^{(2)}) \right|$$
$$\leq \frac{2}{m} M \operatorname{Tr}(\Pi_G D_\ell^{(1)} D_\ell^{(2)} \Pi_G)$$
$$\leq \frac{4}{m} M.$$
By taking union bound, we have with probability at least $1 - \delta$, for all $(x^{(1)}, x^{(2)}) \in \{(x, x), (x, \tilde{x}), (\tilde{x}, \tilde{x})\}$,
$$\left| \frac{2}{m} \frac{b_{\ell+1}^{(1)}}{\sqrt{m}}^\top V_\ell \Pi_G^\perp D_\ell^{(1)} D_\ell^{(2)} \Pi_G^\perp V_\ell^\top \frac{b_{\ell+1}^{(2)}}{\sqrt{m}} - \langle \frac{b_{\ell+1}^{(1)}}{\sqrt{m}}, \frac{b_{\ell+1}^{(1)}}{\sqrt{m}} \rangle \frac{2}{m} \operatorname{Tr}(D_\ell^{(1)} D_\ell^{(2)}) \right|$$
$$\leq \frac{4}{m} \left( \sqrt{2m} M \sqrt{\log \frac{6}{\delta}} + M \log \frac{6}{\delta} \right) + \frac{4}{m} M$$
$$\leq (4 + 4\sqrt{2}) M \sqrt{\frac{1 + \log \frac{6}{\delta}}{m}},$$

where the last inequality holds when $m \geq 1 + \log \frac{6}{\delta}$. $\qquad\square$

**Lemma 12** (Norm controls of $b_{\ell+1}$)**.** *Assume the following inequalities hold simultaneously for all* $\ell = 1, 2, \cdots, L$

$$\left\| \frac{1}{\sqrt{m}} W_\ell \right\| \leq C, \quad \left\| \frac{1}{\sqrt{m}} V_\ell \right\| \leq C.$$

*Then for any fixed input* $x$, $1 \leq \ell \leq L$ *and* $u \in \mathbb{R}^m$, *when*

$$m \geq c L^{2-2\gamma} \log \frac{2L}{\delta'},$$

*with probability at least* $1 - \delta'$ *over the randomness of* $W_{\ell+1}, V_{\ell+1}, \cdots, W_L, V_L, v$, *we have*

$$|\langle u, b_{\ell+1} \rangle| \leq C' \|u\| \sqrt{\log \frac{2L}{\delta'}}.$$

*proof of Lemma 12.* Denote $u_\ell = u$, and

$$u_{i+1} = \alpha \sqrt{\frac{1}{m}} \sqrt{\frac{2}{m}} V_{i+1} D_{i+1} W_{i+1} u_i + u_i, \quad i = \ell, \ell+1, \cdots, L-1.$$

One can show that $\langle u, b_{\ell+1} \rangle = \langle v, u_L \rangle$. Next we show that $\|u_{i+1}\| = (1 + \mathcal{O}(\frac{1}{L}))\|u_i\|$ with high probability. First write

$$\|u_{i+1}\|^2 = \|u_i\|^2 + \alpha^2 \left\| \sqrt{\frac{1}{m}} \sqrt{\frac{2}{m}} V_{i+1} D_{i+1} W_{i+1} u_i \right\|^2 + 2\alpha \left\langle u_i, \sqrt{\frac{1}{m}} \sqrt{\frac{2}{m}} V_{i+1} D_{i+1} W_{i+1} u_i \right\rangle.$$

By the assumption we have

$$\left\| \sqrt{\frac{2}{m}} D_{i+1} W_{i+1} u_i \right\| \leq \sqrt{2} C \|u_i\|,$$

$$\left\| \sqrt{\frac{1}{m}} \sqrt{\frac{2}{m}} V_{i+1} D_{i+1} W_{i+1} u_i \right\| \leq \sqrt{2} C^2 \|u_i\|.$$

With probability at least $1 - \delta'/L$ over the randomness of $V_{i+1}$, we have

$$\left\| \left\langle u_i, \sqrt{\frac{1}{m}} \sqrt{\frac{2}{m}} V_{i+1} D_{i+1} W_{i+1} u_i \right\rangle \right\| \leq \|u_i\| \cdot \left\| \sqrt{\frac{2}{m}} D_{i+1} W_{i+1} u_i \right\| \sqrt{\frac{c \log \frac{2L}{\delta'}}{m}}.$$

Then when

$$m \geq c L^{2-2\gamma} \log \frac{2L}{\delta'},$$

we have

$$\|u_{i+1}\|^2 = \|u_i\|^2 + \alpha^2 \left\| \sqrt{\frac{1}{m}} \sqrt{\frac{2}{m}} V_{i+1} D_{i+1} W_{i+1} u_i \right\|^2 + 2\alpha \langle u_i, \sqrt{\frac{1}{m}} \sqrt{\frac{2}{m}} V_{i+1} D_{i+1} W_{i+1} u_i \rangle$$

$$\leq (1 + 2C^4/L)\|u_i\|^2 + 2\alpha\sqrt{2}C\|u_i\|^2 \sqrt{\frac{c \log \frac{2L}{\delta'}}{m}}$$

$$\leq (1 + 2C^4/L + 2\sqrt{2}C/L)\|u_i\|^2 = (1 + \mathcal{O}(1/L))\|u_i\|^2.$$

Then with probability at least $1 - \delta'(L-1)/L$ we have $\|u_L\| \leq C\|u\|$. Finally the result holds from the standard concentration bound for Gaussian random variables [39]. $\qquad\square$

## C.2 Proofs of Lemma 9

*proof of Lemma 9.* By the assumption, we have

$$\frac{1}{m} \|b_{\ell+1}\|^2 \leq B_{\ell+1}(x, x) + 1 \leq 4.$$

Similarly, $\frac{1}{m}\|\tilde{b}_{\ell+1}\|^2 \leq 4$. Then by Lemma 11, when $m \geq \frac{C}{\epsilon^2}(1 + \log \frac{6}{\delta})$, we have for all $(x^{(1)}, x^{(2)}) \in \{(x, x), (x, \tilde{x}), (\tilde{x}, \tilde{x})\}$,

$$\left| \frac{2}{m} \frac{b_{\ell+1}^{(1)}}{\sqrt{m}}^\top V_\ell \Pi_{\bar{G}}^\perp D_\ell^{(1)} D_\ell^{(2)} \Pi_{\bar{G}}^\perp V_\ell^\top \frac{b_{\ell+1}^{(2)}}{\sqrt{m}} - \langle \frac{b_{\ell+1}^{(1)}}{\sqrt{m}}, \frac{b_{\ell+1}^{(1)}}{\sqrt{m}} \rangle \frac{2}{m} \operatorname{Tr}(D_\ell^{(1)} D_\ell^{(2)}) \right| \leq c\epsilon.$$

Specifically, we have

$$\left\| \sqrt{\frac{2}{m}} \frac{b_{\ell+1}}{\sqrt{m}}^\top V_\ell \Pi_{\bar{G}}^\perp D_\ell \right\| \leq \sqrt{c\epsilon + \frac{2}{m} \operatorname{Tr}(D_\ell) \frac{1}{m} \|b_{\ell+1}\|^2} \leq \mathcal{O}(1),$$

and similarly

$$\left\| \sqrt{\frac{2}{m}} \frac{\tilde{b}_{\ell+1}}{\sqrt{m}}^\top V_\ell \Pi_G^\perp \widetilde{D}_\ell \right\| \le \mathcal{O}(1).$$

Next we bound

$$\left\| \frac{b_{\ell+1}}{\sqrt{m}}^\top V_\ell \Pi_G \right\|.$$

Notice that $\Pi_G$ is a orthogonal projection onto the column space of $G$, which is at most 2-dimension. One can write $\Pi_G = u_1 u_1^\top + u_2 u_2^\top$, where $\|u_i\| = 1$ or $0$. By Lemma 12, fixing $u_1, u_2$ and $V_\ell$, w.p greater than $1 - \delta'$ over the randomness of $W_{\ell+1}, V_{\ell+1}, \cdots, W_L, V_L, v$, we have

$$\left| b_{\ell+1}^\top \frac{1}{\sqrt{m}} V_\ell u_i \right| \le C'' \sqrt{\log \frac{8L}{\delta'}},$$

and

$$\left| \tilde{b}_{\ell+1}^\top \frac{1}{\sqrt{m}} V_\ell u_i \right| \le C'' \sqrt{\log \frac{8L}{\delta'}},$$

for both $i = 1, 2$ when

$$m \ge c L^{2-2\gamma} \log \frac{8L}{\delta'}.$$

Therefore

$$\left\| \frac{b_{\ell+1}}{\sqrt{m}}^\top V_\ell \Pi_G \right\|, \left\| \frac{\tilde{b}_{\ell+1}}{\sqrt{m}}^\top V_\ell \Pi_G \right\| \le \mathcal{O}\left( \sqrt{\log \frac{8L}{\delta'}} \right).$$

Finally, using $I_m = \Pi_G + \Pi_G^\perp$, we have

$$\left| \frac{2}{m} \frac{b_{\ell+1}^{(1)}}{\sqrt{m}}^\top V_\ell D_\ell^{(1)} D_\ell^{(2)} V_\ell^\top \frac{b_{\ell+1}^{(2)}}{\sqrt{m}} - \langle \frac{b_{\ell+1}^{(1)}}{\sqrt{m}}, \frac{b_{\ell+1}^{(2)}}{\sqrt{m}} \rangle \frac{2}{m} \operatorname{Tr}(D_\ell^{(1)} D_\ell^{(2)}) \right|$$

$$\le \left| \frac{2}{m} \frac{b_{\ell+1}^{(1)}}{\sqrt{m}}^\top V_\ell \Pi_G^\perp D_\ell^{(1)} D_\ell^{(2)} \Pi_G^\perp V_\ell^\top \frac{b_{\ell+1}^{(2)}}{\sqrt{m}} - \langle \frac{b_{\ell+1}^{(1)}}{\sqrt{m}}, \frac{b_{\ell+1}^{(1)}}{\sqrt{m}} \rangle \frac{2}{m} \operatorname{Tr}(D_\ell^{(1)} D_\ell^{(2)}) \right|$$

$$+ \sqrt{\frac{2}{m}} \left| \frac{b_{\ell+1}^{(1)}}{\sqrt{m}}^\top V_\ell \Pi_G D_\ell^{(1)} D_\ell^{(2)} \Pi_G^\perp V_\ell^\top \frac{b_{\ell+1}^{(2)}}{\sqrt{m}} \sqrt{\frac{2}{m}} \right|$$

$$+ \sqrt{\frac{2}{m}} \left| \sqrt{\frac{2}{m}} \frac{b_{\ell+1}^{(1)}}{\sqrt{m}}^\top V_\ell \Pi_G^\perp D_\ell^{(1)} D_\ell^{(2)} \Pi_G V_\ell^\top \frac{b_{\ell+1}^{(2)}}{\sqrt{m}} \right|$$

$$+ \frac{2}{m} \left| \frac{b_{\ell+1}^{(1)}}{\sqrt{m}}^\top V_\ell \Pi_G D_\ell^{(1)} D_\ell^{(2)} \Pi_G V_\ell^\top \frac{b_{\ell+1}^{(2)}}{\sqrt{m}} \right|$$

$$\le c\epsilon + \sqrt{\frac{2}{m}} \mathcal{O}\left( \sqrt{\log \frac{8L}{\delta'}} \right) + \frac{2}{m} \mathcal{O}\left( \log \frac{8L}{\delta'} \right) \le \epsilon.$$

The last inequality holds when $m \ge \frac{C}{\epsilon^2} \log \frac{8L}{\delta'}$. $\qquad\qquad\square$

## C.3 Proof of Lemma 10

*proof of Lemma 10.* The first part of the proof is essentially the same as Lemma 9. Define

$$d_{\ell+1} = D_\ell \frac{1}{\sqrt{m}} V_\ell^\top \frac{b_{\ell+1}}{\sqrt{m}}, \quad \tilde{d}_{\ell+1} = \widetilde{D}_\ell \frac{1}{\sqrt{m}} V_\ell^\top \frac{\tilde{b}_{\ell+1}}{\sqrt{m}}.$$

We know that $d_{\ell+1}$ and $\tilde{d}_{\ell+1}$ depend on $W_\ell$ only through $W_\ell x_{\ell-1}$ and $W_\ell \tilde{x}_{\ell-1}$. Let $H = [x_{\ell-1}, \tilde{x}_{\ell-1}]$. Then

$$\left| \frac{2}{m} \langle W_\ell^\top d_{\ell+1}^{(1)}, W_\ell^\top d_{\ell+1}^{(2)} \rangle - 2 \langle d_{\ell+1}^{(1)}, d_{\ell+1}^{(2)} \rangle \right|$$

$$\le \left| \frac{2}{m} \langle \Pi_H^\perp W_\ell^\top d_{\ell+1}^{(1)}, \Pi_H^\perp W_\ell^\top d_{\ell+1}^{(2)} \rangle - 2 \langle d_{\ell+1}^{(1)}, d_{\ell+1}^{(2)} \rangle \right| + \left| \frac{2}{m} \langle \Pi_H W_\ell^\top d_{\ell+1}^{(1)}, \Pi_H W_\ell^\top d_{\ell+1}^{(2)} \rangle \right|$$

$$+ \left| \frac{2}{m} \langle \Pi_H^\perp W_\ell^\top d_{\ell+1}^{(1)}, \Pi_H W_\ell^\top d_{\ell+1}^{(2)} \rangle \right| + \left| \frac{2}{m} \langle \Pi_H W_\ell^\top d_{\ell+1}^{(1)}, \Pi_H W_\ell^\top d_{\ell+1}^{(2)} \rangle \right|.$$

Since $\|d_{\ell+1}\|, \|\tilde{d}_{\ell+1}\| = \mathcal{O}(1)$, similar to Lemma 11, when $m \ge 1 + \log \frac{6}{\delta}$, w.p at least $1 - \delta$ we have

$$\left| \frac{2}{m} \langle \Pi_H^\perp W_\ell^\top d_{\ell+1}^{(1)}, \Pi_H^\perp W_\ell^\top d_{\ell+1}^{(2)} \rangle - 2 \langle d_{\ell+1}^{(1)}, d_{\ell+1}^{(2)} \rangle \right| \le \mathcal{O}\left( \sqrt{\frac{1 + \log \frac{6}{\delta}}{m}} \right),$$

and
$$\left\|\sqrt{\frac{2}{m}}\Pi_H^\perp W_\ell^\top d_{\ell+1}^{(i)}\right\| = \mathcal{O}(1), \quad i = 1, 2,$$
Using the same argument as in the proof of Lemma 9, we decompose $\Pi_H$ into two vectors $w_1$ and $w_2$, whose randomness comes from $W_1, V_1, \cdots, W_{\ell-1}, V_{\ell-1}$. By writing
$$w_i^\top W_\ell^\top d_{\ell+1}^{(i)} = \langle b_{\ell+1}^{(i)}, \frac{1}{\sqrt{m}} V_\ell D_\ell^{(i)} \frac{1}{\sqrt{m}} W_\ell w_i \rangle,$$
we can also apply Lemma 12. Then we conclude that w.p. greater than $1 - \delta'$ over the randomness of $v$, we have
$$\|\Pi_H W_\ell^\top d_{\ell+1}\|, \|\Pi_H W_\ell^\top \tilde{d}_{\ell+1}\| = \mathcal{O}\left(\sqrt{\log \frac{8L}{\delta'}}\right),$$
when
$$m \geq cL^{2-2\gamma} \log \frac{8L}{\delta'}.$$
Then exactly the same result of Lemma 9 holds.

For the second part, notice that
$$\frac{1}{m}\sqrt{\frac{1}{m}}\sqrt{\frac{2}{m}}\langle W_\ell^\top D_\ell^{(1)} V_\ell^\top b_{\ell+1}^{(1)}, b_{\ell+1}^{(2)}\rangle = \sqrt{\frac{2}{m}}\langle W_\ell^\top D_\ell^{(1)} \sqrt{\frac{1}{m}} V_\ell^\top \frac{b_{\ell+1}^{(1)}}{\sqrt{m}}, \frac{b_{\ell+1}^{(2)}}{\sqrt{m}}\rangle$$
$$= \sqrt{\frac{2}{m}}\langle W_\ell^\top d_{\ell+1}^{(1)}, \frac{b_{\ell+1}^{(2)}}{\sqrt{m}}\rangle$$
$$= \sqrt{\frac{2}{m}}\langle \Pi_H^\perp W_\ell^\top d_{\ell+1}^{(1)}, \frac{b_{\ell+1}^{(2)}}{\sqrt{m}}\rangle + \sqrt{\frac{2}{m}}\langle \Pi_H \frac{1}{\sqrt{m}} W_\ell^\top d_{\ell+1}^{(1)}, b_{\ell+1}^{(2)}\rangle.$$
Conditioned on $x_{\ell-1}, \tilde{x}_{\ell-1}, W_\ell x_{\ell-1}$, and $W_\ell \tilde{x}_{\ell-1}$, $W_\ell$ is independent of $b_{\ell+1}, \tilde{b}_{\ell+1}, d_{\ell+1}$, and $\tilde{d}_{\ell+1}$. Furthermore, we have $\Pi_H^\perp W_\ell^\top =_d \Pi_H^\perp \widehat{W}_\ell^\top$, where $\widehat{W}_\ell$ is an i.i.d. copy of $W_\ell$. Then for the first term, with probability at least $1 - \tilde{\delta}/2$, we have for all $(x^{(1)}, x^{(2)}) \in \{(x, x), (x, \tilde{x}), (\tilde{x}, x), (\tilde{x}, \tilde{x})\}$,
$$\left|\sqrt{\frac{2}{m}}\langle \Pi_H^\perp W_\ell^\top d_{\ell+1}^{(1)}, \frac{b_{\ell+1}^{(2)}}{\sqrt{m}}\rangle\right| \leq \left\|\Pi_H^\perp \frac{b_{\ell+1}^{(2)}}{\sqrt{m}}\right\| \|d_{\ell+1}^{(1)}\|\sqrt{\frac{2c \log \frac{16}{\tilde{\delta}}}{m}} \leq \mathcal{O}\left(\sqrt{\frac{\log \frac{16}{\tilde{\delta}}}{m}}\right).$$

For the second term, write $\Pi_H = w_1 w_1^\top + w_2 w_2^\top$, where $\|w_i\| = 1$ or $0$. Then by Lemma 12, with probability at least $1 - \tilde{\delta}/2$, for all $(x^{(1)}, x^{(2)}) \in \{(x, x), (x, \tilde{x}), (\tilde{x}, x), (\tilde{x}, \tilde{x})\}$, when $m \geq cL^{2-2\gamma} \log \frac{16L}{\tilde{\delta}}$, we have
$$\left|\sqrt{\frac{2}{m}}\langle w_i w_i^\top \frac{1}{\sqrt{m}} W_\ell^\top d_{\ell+1}^{(1)}, b_{\ell+1}^{(2)}\rangle\right| = \left|\sqrt{\frac{2}{m}} w_i^\top \frac{1}{\sqrt{m}} W_\ell^\top d_{\ell+1}^{(1)} \langle w_i, b_{\ell+1}^{(2)}\rangle\right|$$
$$\leq \sqrt{\frac{2}{m}}\|w_i\|\left\|\frac{1}{\sqrt{m}} W_\ell^\top\right\|\|d_{\ell+1}^{(1)}\|\left|\langle w_i, b_{\ell+1}^{(2)}\rangle\right|$$
$$\leq \mathcal{O}\left(\sqrt{\frac{\log \frac{16L}{\tilde{\delta}}}{m}}\right).$$
$\square$

## D  Proof of Theorem 5

*Proof.* For $x, \tilde{x} \in \mathbb{S}^{D-1}$, we have $K_\ell(x, x) = K_\ell(\tilde{x}, \tilde{x}) = 1$ for all $\ell$. Hence we only need to study when $x \neq \tilde{x}$. Note we have
$$K_\ell(x, \tilde{x}) = \Gamma_\sigma(K_{\ell-1})(x, \tilde{x}) = \hat{\sigma}(K_{\ell-1}(x, \tilde{x})), \text{ and } \Gamma_{\sigma'}(K_\ell)(x, \tilde{x}) = \widehat{\sigma'}(K_\ell(x, \tilde{x})).$$
For simplicity, we use $K_\ell$ to denote $K_\ell(x, \tilde{x})$, where $x \neq \tilde{x}$ and $x, \tilde{x} \in \mathbb{S}^{D-1}$.

Recall that
$$\hat{\sigma}(\rho) = \frac{\sqrt{1 - \rho^2} + (\pi - \cos^{-1}(\rho)) \rho}{\pi}, \text{ and } \widehat{\sigma'}(\rho) = \frac{\pi - \cos^{-1}(\rho)}{\pi}.$$
Hence we have $\hat{\sigma}(1) = 1$, $K_{\ell-1} \leq \hat{\sigma}(K_{\ell-1}) = K_\ell$, $(\hat{\sigma})'(\rho) = \widehat{\sigma'}(\rho) \in [0, 1]$, and $(\widehat{\sigma'})'(\rho) \geq 0$. Then $\hat{\sigma}$ is a convex function.

Since $\{K_\ell\}$ is an increasing sequence and $|K_\ell| \leq 1$, we have $K_\ell$ converges as $\ell \to \infty$. Taking the limit of both sides of $\hat{\sigma}(K_{\ell-1}) = K_\ell$, we have $K_\ell \to 1$ as $\ell \to \infty$.

For $K_\ell$, we also have

$$K_\ell = \hat{\sigma}(K_{\ell-1}) = \frac{\sqrt{1 - K_{\ell-1}^2} + (\pi - \cos^{-1}(K_{\ell-1}))K_{\ell-1}}{\pi} = K_{\ell-1} + \frac{\sqrt{1 - K_{\ell-1}^2} - \cos^{-1}(K_{\ell-1})K_{\ell-1}}{\pi}.$$

Let $e_\ell = 1 - K_\ell$, we can easily check that

$$e_{\ell-1} - \frac{e_{\ell-1}^{3/2}}{\pi} \leq e_\ell \leq e_{\ell-1} - \frac{2\sqrt{2}e_{\ell-1}^{3/2}}{3\pi}. \tag{19}$$

Hence as $e_\ell \to 0$, we have $\frac{e_\ell}{e_{\ell-1}} \to 1$, which implies $\{K_\ell\}$ converges sublinearly.

Assume $e_\ell = \frac{C}{\ell^p} + \mathcal{O}(\ell^{-(p+1)})$. By taking the assumption into (19) and comparing the highest order of both sides, we have $p = 2$.

Thus $\exists C$, s.t. $|1 - K_\ell| \leq \frac{C}{\ell^2}$, i.e. the convergence rate of $K_\ell$ is $\mathcal{O}\left(\frac{1}{\ell^2}\right)$.

**Lemma 13.** *For each $K_0 < 1$, there exists $p > 0$ and $n_0 = n_0(\delta) > 0$, such that $K_n \leq 1 - \frac{9\pi^2}{2(n+n_0)^{2+\frac{\log(L)^p}{L}}}$, $\forall n = 0, \ldots, L$, when $L$ is large.*

*Proof.* First, solve $K_0 \leq 1 - \frac{9\pi^2}{2n^{2+\frac{\log(L)^p}{L}}}$. Then we can choose $n_0 \geq \sqrt{\frac{9\pi^2}{2\delta}} \geq \sqrt{\frac{9\pi^2}{2(1-K_0)}}$, which is independent of $L$ and $n$. For the rest of the proof, without loss of generality, we just use $n$ instead of $n + n_0$. Also for small $\delta($ when $\delta$ is not small enough we can pick a small $\delta_0 < \delta$ and let $n_0 \geq \sqrt{\frac{9\pi^2}{2\delta_0}})$, we have $\frac{9\pi^2}{2(n+n_0)^{2+\frac{\log(L)^p}{L}}} \leq \delta$(or $\delta_0$) which is also small.

Let $K_n = 1 - \epsilon$. Then, when $\epsilon$ is small, we have
$$K_{n+1} - K_n = \hat{\sigma}(K_n) - K_n = \mathcal{O}(\epsilon^{3/2}).$$

Also, we have

$$\left(1 - \frac{9\pi^2}{2(n+1)^{2+\frac{\log(L)^p}{L}}}\right) - \left(1 - \frac{9\pi^2}{2n^{2+\frac{\log(L)^p}{L}}}\right) = \mathcal{O}\left(\frac{1}{n^{3+\frac{\log(L)^p}{L}}}\right)$$

$$\geq \mathcal{O}\left(\left(\frac{1}{n^{2+\frac{\log(L)^p}{L}}}\right)^{3/2}\right) = \mathcal{O}\left(\frac{1}{n^{3+\frac{3\log(L)^p}{2L}}}\right).$$

Overall, we want an upper bound for $K_n$ and from the above we only know that $K_n$ is of order $1 - \mathcal{O}(n^{-2})$ but this order may hide some terms of logarithmic order. Hence we use the order $1 - \mathcal{O}(n^{-(2+\epsilon)})$ to provide an upper bound of $K_n$. Here $\frac{\log(L)^p}{L}$ is constructed for the convenience of the rest of the proof. $\square$

Let $N_0 = N_0(L)$ be the solution of

$$\cos\left(\pi\left(1 - \left(\frac{n+1}{n+2}\right)^{3-\frac{\log(L)^2}{L}}\right)\right) = \hat{\sigma}\left(\cos\left(\pi\left(1 - \left(\frac{n}{n+1}\right)^{3-\frac{\log(L)^2}{L}}\right)\right)\right),$$

where for $N_0 < n < N_L$ with some $N_L$, we have

$$\cos\left(\pi\left(1 - \left(\frac{n+1}{n+2}\right)^{3-\frac{\log(L)^2}{L}}\right)\right) \geq \hat{\sigma}\left(\cos\left(\pi\left(1 - \left(\frac{n}{n+1}\right)^{3-\frac{\log(L)^2}{L}}\right)\right)\right).$$

One can check by series expansion that $N_0 = N_0(L) \leq 5\frac{L}{\log(L)^2}$.

Next we would like to find $n$ such that

$$K_n = \cos\left(\pi\left(1 - \left(\frac{5\frac{L}{\log(L)^2}}{5\frac{L}{\log(L)^2} + 1}\right)^{3-\frac{\log(L)^2}{L}}\right)\right).$$

By series expansion, we know

$$\cos\left(\pi\left(1-\left(\frac{5\frac{L}{\log(L)^2}}{5\frac{L}{\log(L)^2}+1}\right)^{3-\frac{\log(L)^2}{L}}\right)\right)\geq 1-\frac{9\pi^2}{2\left(\frac{5L}{\log(L)^2}\right)^2}.$$

Then it suffices to solve

$$1-\frac{9\pi^2}{2(\frac{5L}{\log(L)^2})^2}\geq 1-\frac{9\pi^2}{2n^{2+\frac{\log(L)^p}{L}}}\geq K_n,\ i.e.,\ n^{2+\frac{\log(L)^p}{L}}\leq\left(\frac{5L}{\log(L)^2}\right)^2. \tag{20}$$

**Lemma 14.** *When $q > p-1$, we have $n \lesssim \frac{5L}{\log(L)^2}-\log(L)^q$ satisfies (20).*

*Proof.* If the condition above holds, we have

$$n^{2+\frac{\log(L)^p}{L}}\leq\left(\frac{5L}{\log(L)^2}-\log(L)^q\right)^{2+\frac{\log(L)^p}{L}},$$

which is

$$n^{1+\frac{\log(L)^p}{2L}}\leq\left(\frac{5L}{\log(L)^2}-\log(L)^q\right)\left(\frac{5L}{\log(L)^2}-\log(L)^q\right)^{\frac{\log(L)^p}{2L}}$$

$$\leq\left(\frac{5L}{\log(L)^2}-\log(L)^q\right)\left(1+\frac{\log(L)^p\log(\frac{5L}{\log(L)^2})}{2L}\right)$$

$$=\frac{5L}{\log(L)^2}-\log(L)^q+\frac{5}{2}\log(L)^{p-2}\log\left(\frac{5L}{\log(L)^2}\right)-\frac{1}{2L}\log(L)^{p+q}\log\left(\frac{5L}{\log(L)^2}\right),$$

where $\left(\frac{5L}{\log(L)^2}-\log(L)^q\right)^{\frac{\log(L)^p}{2L}}\to 1$ as $L\to\infty$.

Thus we have $q > p-1$. $\qquad\square$

Just pick $q = p$. Then we have $n^{1+\frac{\log(L)^p}{2L}}\lesssim\frac{5L}{\log(L)^2}$ and $n\lesssim\frac{5L}{\log(L)^2}-\log(L)^p$.

**Lemma 15.** *When $L$ is large enough, we have*

$$\cos\left(\pi\left(1-\left(\frac{n}{n+1}\right)^{3+\frac{\log(L)^2}{L}}\right)\right)\leq K_n\leq\cos\left(\pi\left(1-\left(\frac{n+\log(L)^p}{n+\log(L)^p+1}\right)^{3-\frac{\log(L)^2}{L}}\right)\right).$$

*Proof.* Let $F(n)=\cos\left(\pi\left(1-\left(\frac{n+\log(L)^p}{n+\log(L)^p+1}\right)^{3-\frac{\log(L)^p}{L}}\right)\right).$

For the right hand side, when $n\gtrsim\frac{5L}{\log(L)^2}-\log(L)^p$, we have, by series expansion, $F(n+1)\geq\hat\sigma\left(F(n)\right)$. Also, when $n\sim aL$, where $0<a\leq 1$, we have

$$F(n+1)-\hat\sigma(F(n))=\mathcal{O}\left(\frac{3\left(2\pi^2\mathrm{a}\log^{10}(L)+\pi^2\log^8(L)\right)}{2L^4\left(\mathrm{a}\log^2(L)+5\right)^4}\right)>0.$$

Then for $\frac{5L}{\log(L)^2}-\log(L)^p\lesssim n\lesssim L$, we have $F(n+1)\geq\hat\sigma\left(F(n)\right)$ and thus $K_n\leq F(n)$.

When $n\lesssim\frac{5L}{\log(L)^2}-\log(L)^p$, we have $F(n+1)\leq\hat\sigma\left(F(n)\right)$. Hence $K_n\leq F(n)$.

For the left hand side,

$$\cos\left(\pi\left(1-\left(\frac{n+1}{n+2}\right)^{3+\frac{\log(L)^2}{L}}\right)\right)-\hat\sigma\left(\cos\left(\pi\left(1-\left(\frac{n}{n+1}\right)^{3+\frac{\log(L)^2}{L}}\right)\right)\right)$$

$$\sim-\frac{27\pi^2}{2n^4}-\frac{3\pi^2\log(L)^2}{n^3L},\ \forall n=1,...,L.$$

Hence we have the left hand side. $\qquad\square$

From Lemma 15, by series expansion, we have

$$|1 - K_n| \leq \frac{\left(3\pi + \frac{\pi \log(L)^2}{L}\right)^2}{2n^2} \sim \frac{9\pi^2}{2n^2},$$

when $L$ is large.

Moreover, we can get

$$\left(\frac{n}{n+1}\right)^{3+\frac{\log(L)^2}{L}} \leq \Gamma_{\sigma'}(K_n) \leq \left(\frac{n + \log(L)^p}{n + \log(L)^p + 1}\right)^{3-\frac{\log(L)^2}{L}}.$$

Then

$$\left(\frac{\ell-1}{L}\right)^{3+\frac{\log(L)^2}{L}} \leq \prod_{i=\ell}^{L} \Gamma_{\sigma'}(K_{i-1}) \leq \left(\frac{\ell + \log(L)^p - 1}{L + \log(L)^p}\right)^{3-\frac{\log(L)^2}{L}}.$$

Let $N = \log(L)^p$. For the right hand side, if we sum over $\ell$, we have

$$\frac{1}{L}\sum_{\ell=1}^{L}\left(\frac{\ell + N - 1}{L + N}\right)^{3-\frac{\log(L)^2}{L}} \leq \frac{1}{L}\int_1^{L+1}\left(\frac{x + N - 1}{L + N}\right)^{3-\frac{\log(L)^2}{L}} dx$$

$$= \frac{\left((L+N)^{4-\frac{\log(L)^2}{L}} - (N)^{4-\frac{\log(L)^2}{L}}\right)}{L(L+N)^{3-\frac{\log(L)^2}{L}}\left(4 - \frac{\log(L)^2}{L}\right)}.$$

Taking the limit of both sides, we have

$$\lim_{L\to\infty} \frac{1}{L}\sum_{\ell=1}^{L}\left(\frac{\ell + N - 1}{L + N}\right)^{3-\frac{\log(L)^2}{L}} \leq \frac{1}{4}.$$

Similarly, by

$$\frac{1}{L}\sum_{i=1}^{L}\left(\frac{\ell-1}{L}\right)^{3+\frac{\log(L)^2}{L}} \geq \frac{1}{L}\int_1^{L}\left(\frac{x-1}{L}\right)^{3+\frac{\log(L)^2}{L}} dx = \frac{(L-1)^{4+\frac{\log(L)^2}{L}}}{\left(4 + \frac{\log(L)^2}{L}\right)L^{4+\frac{\log(L)^2}{L}}},$$

we have

$$\lim_{L\to\infty}\frac{1}{L}\sum_{i=1}^{L}\left(\frac{\ell-1}{L}\right)^{3+\frac{\log(L)^2}{L}} \geq \frac{1}{4}.$$

Hence,

$$\lim_{L\to\infty}\frac{1}{L}\sum_{\ell=1}^{L}\left(\frac{\ell + N - 1}{L + N}\right)^{3-\frac{\log(L)^2}{L}} = \lim_{L\to\infty}\frac{1}{L}\sum_{\ell=1}^{L}\left(\frac{\ell-1}{L}\right)^{3+\frac{\log(L)^2}{L}}$$

$$= \lim_{L\to\infty}\frac{1}{L}\sum_{\ell=1}^{L}\prod_{i=\ell}^{L}\Gamma_{\sigma'}(K_{i-1}) = \frac{1}{4}.$$

Recall from previous discussion, $K_\ell = 1 - \mathcal{O}(\frac{1}{\ell^2})$. Therefore,

$$\lim_{L\to\infty}\frac{1}{L}\sum_{\ell=1}^{L} K_{\ell-1}\prod_{i=\ell}^{L}\Gamma_{\sigma'}(K_{i-1}) = \frac{1}{4}.$$

Also, when $L$ is large, we have

$$\frac{\left((L+N)^{4-\frac{\log(L)^2}{L}} - (N)^{4-\frac{\log(L)^2}{L}}\right)}{L(L+N)^{3-\frac{\log(L)^2}{L}}\left(4 - \frac{\log(L)^2}{L}\right)} > \frac{1}{4} > \frac{(L-1)^{4+\frac{\log(L)^2}{L}}}{\left(4 + \frac{\log(L)^2}{L}\right)L^{4+\frac{\log(L)^2}{L}}}.$$

Hence we can estimate the convergence rate of the normalized kernel

$$\left|\frac{1}{L}\sum_{\ell=1}^{L}K_{\ell-1}\prod_{i=\ell}^{L}\Gamma_{\sigma'}(K_{i-1}) - \frac{1}{4}\right| = \left|\frac{1}{L}\sum_{\ell=1}^{L}\left(K_{\ell-1}\left(\prod_{i=\ell}^{L}\Gamma_{\sigma'}(K_{i-1}) - \frac{1}{4}\right) + \frac{1}{4}(K_{\ell-1} - 1)\right)\right|$$

$$\leq \left| \frac{1}{L}\sum_{\ell=1}^{L}\prod_{i=\ell}^{L}\Gamma_{\sigma'}(K_{i-1}) - \frac{1}{4} \right| + \frac{1}{4}\left| \frac{1}{L}\sum_{\ell=1}^{L}(K_{\ell-1}-1) \right|$$

$$\leq \left| \frac{\left((L+N)^{4-\frac{\log(L)^2}{L}} - (N)^{4-\frac{\log(L)^2}{L}}\right)}{L(L+N)^{3-\frac{\log(L)^2}{L}}\left(4-\frac{\log(L)^2}{L}\right)} - \frac{(L-1)^{4+\frac{\log(L)^2}{L}}}{\left(4+\frac{\log(L)^2}{L}\right)L^{4+\frac{\log(L)^2}{L}}} \right|$$

$$+ \frac{1}{4}\left| \frac{1}{L}\sum_{i=1}^{L}(K_{\ell-1}-1) \right|$$

$$\lesssim \frac{4\log(L)^p + \log(L)^2}{16L} = \mathcal{O}\left( \frac{\text{poly}\log(L))}{L} \right)$$

$$\square$$

## E  Proof of Theorem 6

*Proof.* We denote $K_{\ell,L}$ to be the $\ell$-th layer of $K$ when the depth is $L$, which is originally denoted by $K_\ell$.

Let $S_{\ell,L} = \frac{K_{\ell,L}}{(1+\alpha^2)^\ell} = \frac{K_{\ell,L}}{(1+1/L^2)^\ell}$ and $S_0 = K_0$, then $\Gamma_\sigma(K_{\ell,L}) = (1+\alpha^2)^\ell \hat{\sigma}(S_{\ell,L})$ and $\Gamma_{\sigma'}(K_{\ell,L}) = \widehat{\sigma'}(S_{\ell,L})$. Hence we can rewrite the recursion to be

$$S_{\ell,L} = \frac{S_{\ell-1,L} + \alpha^2 \hat{\sigma}(S_{\ell-1,L})}{(1+\alpha^2)} \geq S_{\ell-1,L}. \tag{21}$$

Moreover, since $S_{\ell,L}-S_{\ell-1,L} = \frac{\alpha^2}{1+\alpha^2}(\hat{\sigma}(S_{\ell-1,L})-S_{\ell-1,L})$ and $(\hat{\sigma}(S_{\ell-1,L})-S_{\ell-1,L})$ is decreasing, we can have

$$S_{\ell,L} \leq S_0 + \frac{(\hat{\sigma}(S_0)-S_0)\ell}{L^2}.$$

Denote $P_{\ell+1,L} = B_{\ell+1,L}(1+\alpha^2)^{-(L-\ell)} = \prod_{i=\ell}^{L-1}\frac{1+\alpha^2\widehat{\sigma'}(S_{i,L})}{1+\alpha^2}$. Since

$$1 - \frac{1+\alpha^2\widehat{\sigma'}(S_{i,L})}{1+\alpha^2} = \frac{\alpha^2(1-\widehat{\sigma'}(S_{i,L}))}{1+\alpha^2} = \frac{1-\widehat{\sigma'}(S_{i,L})}{L^2+1},$$

we have

$$1 - P_{\ell+1,L} = 1 - \prod_{i=\ell}^{L-1}\left(1 - \frac{1-\widehat{\sigma'}(S_{i,L})}{L^2+1}\right) \leq \sum_{i=\ell}^{L-1}\frac{1-\widehat{\sigma'}(S_{i,L})}{L^2+1} = \frac{L-\ell-\sum_{i=\ell}^{L-1}\widehat{\sigma'}(S_{i,L})}{L^2+1},$$

where $\ell = 1,\ldots,L-1$. For $P_{L+1,L}$, we have $1 - P_{L+1,L} = 0$.

Then we can rewrite the normalized kernel to be

$$\overline{\Omega}_L = \frac{1}{2L}\sum_{\ell=1}^{L}P_{\ell+1,L}(\hat{\sigma}(S_{\ell-1,L}) + S_{\ell-1,L}\widehat{\sigma'}(S_{\ell-1,L})).$$

Hence we have the bound for each layer

$$\left| P_{\ell+1,L}(\hat{\sigma}(S_{\ell-1,L}) + S_{\ell-1,L}\widehat{\sigma'}(S_{\ell-1,L})) - (\hat{\sigma}(S_0) + S_0\widehat{\sigma'}(S_0)) \right|$$

$$\leq \left| P_{\ell+1,L} \right| \cdot \left| (\hat{\sigma}(S_{\ell-1,L}) + S_{\ell-1,L}\widehat{\sigma'}(S_{\ell-1,L})) - (\hat{\sigma}(S_0) + S_0\widehat{\sigma'}(S_0)) \right| + \left| \hat{\sigma}(S_0) + S_0\widehat{\sigma'}(S_0) \right| \cdot \left| 1 - P_{\ell+1,L} \right|$$

$$\leq \left| \widehat{\sigma'}(S_{\ell-1,L})(S_{\ell-1,L} - S_0) \right| + \left| \widehat{\sigma'}(S_{\ell-1,L})S_{\ell-1,L} - \widehat{\sigma'}(S_0)S_0 \right| + \left| \hat{\sigma}(S_0) + S_0\widehat{\sigma'}(S_0) \right| \cdot \left| 1 - P_{\ell+1,L} \right|$$

$$= 2\left| \widehat{\sigma'}(S_{\ell-1,L})(S_{\ell-1,L} - S_0) \right| + \left| S_0(\widehat{\sigma'}(S_{\ell-1,L}) - \widehat{\sigma'}(S_0)) \right| + \left| \hat{\sigma}(S_0) + S_0\widehat{\sigma'}(S_0) \right| \cdot \left| 1 - P_{\ell+1,L} \right|$$

$$\leq \frac{2\widehat{\sigma'}(S_{\ell-1,L})(\hat{\sigma}(S_0) - S_0)\ell}{L^2} + \frac{|S_0|(\hat{\sigma}(S_0) - S_0)(\ell-1)}{\pi L^2\sqrt{1-S_{\ell-1,L}^2}} + \left| \hat{\sigma}(S_0) + S_0\widehat{\sigma'}(S_0) \right|\frac{L-\ell-\sum_{i=\ell}^{L-1}\widehat{\sigma'}(S_{i,L})}{L^2+1}$$

$$\leq \frac{2\widehat{\sigma'}(S_{\ell-1,L})(\hat{\sigma}(S_0) - S_0)\ell}{L^2} + \frac{|S_0|(\hat{\sigma}(S_0) - S_0)(\ell-1)}{\pi L^2\sqrt{1-S_{\ell-1,L}^2}} + \left| \hat{\sigma}(S_0) + S_0\widehat{\sigma'}(S_0) \right|\frac{L-\ell-(L-\ell)\widehat{\sigma'}(S_0)}{L^2+1}.$$

Therefore we have the bound for the normalized kernel

$$\left| \overline{\Omega}_L - \frac{1}{2}(\hat{\sigma}(S_0) + S_0\widehat{\sigma'}(S_0)) \right|$$

$$= \left| \frac{1}{2L} \sum_{\ell=1}^{L} \left( P_{\ell+1,L}(\hat{\sigma}(S_{\ell-1,L}) + S_{\ell-1,L}\widehat{\sigma'}(S_{\ell-1,L})) \right) - \frac{1}{2}(\hat{\sigma}(S_0) + S_0\widehat{\sigma'}(S_0)) \right|$$

$$\leq \frac{1}{2L} \sum_{\ell=1}^{L} \left( \frac{2\widehat{\sigma'}(S_{\ell-1,L})(\hat{\sigma}(S_0) - S_0)\ell}{L^2} + \frac{|S_0|(\hat{\sigma}(S_0) - S_0)(\ell-1)}{\pi L^2 \sqrt{1 - S_{\ell-1,L}^2}} \right)$$

$$+ \frac{1}{2L} \sum_{\ell=1}^{L-1} \left( \left| \hat{\sigma}(S_0) + S_0\widehat{\sigma'}(S_0) \right| \frac{L - \ell - (L-\ell)\widehat{\sigma'}(S_0)}{L^2 + 1} \right)$$

$$\leq \frac{1}{2L} \left( \frac{L+1}{L}(\hat{\sigma}(S_0) - S_0) + \frac{|S_0|(\hat{\sigma}(S_0) - S_0)L(L-1)}{2\pi L^2 C} + \left| \hat{\sigma}(S_0) + S_0\widehat{\sigma'}(S_0) \right| \frac{\frac{L(L-1)}{2}(1 - \widehat{\sigma'}(S_0))}{L^2 + 1} \right)$$

$$\sim \left( \frac{(\hat{\sigma}(S_0) - S_0)}{2} \left( 1 + \frac{|S_0|}{2\pi C} \right) + \frac{1}{2} \left| \hat{\sigma}(S_0) + S_0\widehat{\sigma'}(S_0) \right| (1 - \widehat{\sigma'}(S_0)) \right) \frac{1}{L}$$

where $C = C(\delta) = \sqrt{1 - (1-\delta)^2}$ and $S_0 = K_0$. $\square$