[Reviews · NeurIPS 2020]

Review 1

Summary and Contributions: This paper computes the Gaussian process kernel (GPK) and the neural tangent kernel (NTK) of a specific ResNet-like architectures, which characterize the limiting behavior of NNs when the size m of their hidden layers becomes infinite, and when training only the last network layer (GPK case) or when training all layers but the last and first ones (NTK case). The authors then compute the limit NTK of ResNets when their number of layers L grows indefinitely, and compare it to the limit NTK of classical feedforward nets (FFNets). They find that, while the limit NTK of FFNets is a degenerate discrete kernel (constant + nugget in 0) --which can fit any data but cannot generalize--, the limit NTK of ResNets is smooth and close --and sometimes equal-- to the NTK of a ResNet of depth 1. To the authors, this explains why learning with FFNets is limited to a few dozen layers only while ResNets have thousands, and why adding ResNet layers changes the accuracy only marginally. The authors however admit (and confirm empirically) that in practice, even for infinitely wide ResNets, the number of layers does slightly influence the performance. Finite sample guarantees are provided for all convergence results.

Strengths: The article provides an interesting analysis and comparison of the limiting behavior of infinitely wide FFNets and ResNets when their depth increases, which brings us a step closer towards understanding the behavior and compared generalization abilities of some SOTA network architecture (FFNets vs ResNets).

Weaknesses: The essential parts of the proofs are in appendix, which reviewers are not required to check, and which would anyway be too long (17 pages) for me to review in such a short period. Therefore, all propositions and theorems should at least be validated/double-checked by empirical simulations included in the paper, f.ex. by letting neural networks become sufficiently large and deep, and checking that they behave as predicted by the theorems. Please note that these experiments would be different from those of Figs. 2 & 3. What I ask for is a simple, yet strong safety check, that's easy to review and confirms the validity of the theorems. I do understand that, mathematically speaking, a proven result does not need confirmation. Still, in practice, and given the reviewing circumstances, such safety checks would be useful/critical both for the authors and the NeurIPS reviewing committee, who, by accepting the paper, would endorse its correctness and engage its own credibility. ### Minor points: - The paper only analyses fully connected ResNets, whereas ResNets typically use convolutional layers in practice. The paper should at least clearly state this obvious limitation and include it in the Broader Impact section. - The paper only analyzes a specific scaling of the bottleneck weight $\alpha = L^\gamma$, without further justification. I don't recall seeing such a scaling in any applied work. Therefore, a small justification would be the least. - Practitioners systematically include batchnorms in ResNets (and FFNets), which are not taking into account by this paper. Therefore, it would be interesting to add some experiments to study whether these batchnorms actually alter the predictions of the theorems. (These batchnorm experiments could be coupled with or compared to the safety-check experiments mentioned above.)

Correctness: Results seem plausible, but since I did not check the proofs (17 pages in appendix), and since the paper did not include any "safety check" experiments to confirm the theorems, it's impossible to say for sure. See "Weaknesses".

Clarity: The paper is very clear and well written.

Relation to Prior Work: Yes, well discussed.

Reproducibility: Yes

Additional Feedback: ### On my overall decision I am willing to largely upgrade my decision, if the authors can provide strong evidence that's easy to check (i.e. "safety checks") to support the correctness of their propositions/theorems. ### Limit NTK, ResNet-1 and unrolled recurrent networks That the limit NTK of a ResNet is the NTK of a ResNet-1 may seem very surprising, but I wonder if the following reasoning could serve as an easy intuitive confirmation: ResNets are essentially recurrent neural networks that got unrolled over L time-steps and with time-varying weights. But since the size m of the hidden layers becomes infinite, the set of weights tends to a fixed limiting distribution: the same for all layers. Therefore, when m goes to infinity, the time-varying component gets smoothed out. So, when L now becomes infinite, we exactly recover an unrolled, 1-layer recurrent neural network. ### Typos - l.201: notion -> fashion ? - l.241: By Representer theorem -> By the representer theorem - Fig.2, caption: CIFAR102 -> CIFAR2 Reply to author response ----------------------------- Thank you for the additional plots provided in your response, which indeed nicely confirm your main theorems. Please include them in your final version (at least in appendix). I'll increase my score accordingly. Minor remarks: - Please add error bars (f.ex. 5-95 percentiles) to fig. (a) of your response: Thm.4 claims that the empirical and theoretical inner products should be close with high probability (not just in expectation), meaning that even the 5-95 percentiles should be close to the theoretical values (for deep and wide enough nets). - About conv-layers and BN: I do believe that your techniques can be applied to conv-layers, but I would be very interested to see how you'd apply it to BN. Could you add a small comment on this in your final version (possibly in appendix)? - About the scaling factor: just saying that your assumption is common doesn't mean it's sensible... I'd be happy to see a small, clear justification for it.


Review 2

Summary and Contributions: This paper calculates the equivalent Gauss process kernel and the equivalent neural tangent kernel of any ResNet with an infinite width. It then proves that when the depth of a ResNet goes to infinity, the equivalent kernels of this ResNet converges to a one-layer network. The authors then conducted experiments to verify these results.

Strengths: The strengths are listed below. 1. This result is impressive: when the depth of a ResNet goes to infinity, the equivalent kernels of this ResNet converges to a one-layer network. It may shed lights to fully understanding ResNets’ behaviour. It also suggests a degenerative nature of the deep leaning models. 2. The proofs are solid and given in detail.

Weaknesses: My concerns are listed as follows. 1. The assumptions are still quite strong: the width needs to be infinite. There still exists a gap between this paper and the practice. It is desired to give an "error analyse" for the “distance” between networks in such an overparameterized regime and those used in practice. For example, how significant would the results change if the width of the network is bounded? 2. The authors suggest that its theoretical results (“infinite-depth networks are equivalent to one-layer networks”) supports ResNets’ good generalizability. A direct measure for generalizability would be a generalization bound (or hypothesis complexity here). Would the current result help deliver a smaller generalization bound? Or, does the current form of equivalence (in the view of NTK) straightforwardly lead to the equivalence in the view of complexity or generalizability? ====== Thanks for the response. My concerns are cleared. It would be great to add the explanation regarding generalization in the next version.

Correctness: The claims, method, and empirical methodology are correct.

Clarity: This paper is well-written.

Relation to Prior Work: The differences with previous contributions are clearly discussed.

Reproducibility: Yes

Additional Feedback: It is desired to clarify the two aforementioned weaknesses in the rebuttal session.


Review 3

Summary and Contributions: The authors show differences between the limiting kernel for neural networks when depth goes to infinite for the NTK induced by a standard feed forward network and a residual network. In particular in their framework they show that the NTK for an infinite depth feedforward network is degenerate, whereas of a residual network the infinite depth limit is not.

Strengths: The relationship between kernels and neural networks is a timely and important problem. Showing that the infinite depth NTK for a feedforward network is degenerate is somewhat interesting.

Weaknesses: Post Rebuttal Update: While I think the paper is still below the bar for acceptance (as I think the motivation is weak) the rebuttal and discussion with other reviewers on the novelty of the theoretical work has encouraged me to increase my score to a 5. Overall this paper is poorly motivated and written. It is unclear why taking depth to infinite is the correct regime to study when networks that achieve state of the art accuracy on ImageNet or Cifar are not necessarily the deepest (50 layers for cifar, few 100 for ImageNet). Thus the whole motivation for the paper is somewhat questionable. The experiments are very weak and use non-standard 2 class versions of MNIST and CIFAR. It is unclear why the authors do not compare to published kernel results on CIFAR-10 and MNIST. I think a cleaner rewrite of the theoretical result and stronger experimental section with rigorous experimentation that shows *exactly at what point* deeper feedforward kernels degrade and compare this to when deeper feedforward neural networks degrade could make a significantly stronger paper.

Correctness: I did not verify the correctness of the theorems but I can buy they are true, and the experiments seem weak but correct.

Clarity: No, there are 5 theorems in the paper and it is unclear what point they serve. The entire paper reads like a wall of lemmas and theorems and it is utterly unclear the motivation for each mathematical statement.

Relation to Prior Work: The paper doesn't cite much of the relevant prior work. Greg Yang has derivations for neural tangent kernels for various architecture: https://arxiv.org/abs/2006.14548 Shankar et al show (contrary to the results in this paper) that for moderate depth increases test accuracy does improve for neural kernels (NNGP not NTK): https://arxiv.org/abs/2003.02237

Reproducibility: Yes

Additional Feedback:


Review 4

Summary and Contributions: Consider a (variant of the) ResNet, and draw its weights independently from a Gaussian with zero mean, and appropriate variance that is inversely proportional to the width. This paper proves that the L2-norm squared of the pre-activations of each layer of a finite ResNet, with finite (but large enough) width, is close to the ResNet's corresponding GP kernel in a probably-approximately sense. This is stronger than what was shown by [Y19], and similar to what was shown by [27, 14]. Theorem 3 tells us about how close to the NN-GP kernel the networks with finite width are, as opposed to talking about almost-sure convergence in the limit (which is [Y19]). The authors prove the same for the inner product of the gradient other all parameters, and the ResNet's corresponding NTK (Theorem 4). The authors then use these proofs to link the following statement about the NTK of a feed-forward network and a ResNet, to the behaviour of the actual networks: As the depth of the network goes to infinity, the feed-forward network's NTK forgets about what the two inputs were. With some Tikhonov regularization, the kernel can fit any finite training set; however, the predictions outside of the training set are always 0. It follows that the concept class of this kernel is not learnable, that is, it is not the case that for every learning distribution and function x->y, the kernel machine can identify x->y from a finite training set. However, the ResNet kernel does not forget about the input with infinite depth, and so its concept class is learnable. This partially explains why ResNets generalize better than FFNets in deep settings. [Y19] Yang, Greg. https://papers.nips.cc/paper/9186-wide-feedforward-or-recurrent-neural-networks-of-any-architecture-are-gaussian-processes [27] Daniely, A., Frostig, R., & Singer, Y., Toward deeper understanding of neural networks: the power of initialization and a dual view on expressivity. [14] Allen-Zhu, Z., Li, Y., & Song, Z., A convergence theory for deep learning via over-parameterization,

Strengths: The paper provides a mathematically rigorous and sound treatment of (a variant of) ResNets. The proofs presented here are novel. That infinitely deep ResNets do not forget about the input, but finite ones do and therefore cannot learn, is an important result. It is rigorously proven here. The last result presented is verified numerically. A related result (with gamma=0.5) is also numerically tested, though not proven.

Weaknesses: I'm satisfied with the rebuttal. The authors did a good job. ==== The central conclusion of this work, that infinitely deep FFNets cannot learn whereas ResNets can, is non-rigorously known, though not written in a single paper. Although the authors do put it in a firmer ground, I believe they should acknowledge this. More broadly, I think we have a fairly good idea about what separates the performance of ResNets from FFNets. In addition to the previous paragraph, the "edge of chaos" papers ([4] and the previous one on FFNets) provide a fairly good explanation in terms of gradient vanishing or lack thereof in the case of FFNets and ResNets. The introduction does not give this impression. Changing the framing of the paper so that these things are emphasised would make me more confident in recommending it for acceptance. ---- Radford Neal already wrote in his 1996 book "Bayesian Learning for Neural Networks" (page 50-51): "Finally, we can consider the limiting behaviour of the prior over functions as the number of hidden layers increases. If the priors [...] are the same for all hidden layers, the prior over the functions computed by the units in the hidden layers of such a network will have the form of a homogeneous Markov chain. We can now ask whether this Markov chain converges to some invariant distribution as the number of layers goes to infinity, [...]. If the chain does converge, then the prior over functions computed by the output units should also converge, since the outputs are computed solely from the hidden units in the last layer. [...] from the discussion above, it is clear that a Gaussian-based prior for a network with many layers of step-function hidden units, with no direct connections from inputs to hidden layers after the first, either does not converge as the number of layers goes to infinity, or if it can be regarded as converging, it is to an uninteresting distribution concentrated on completely unlearnable functions." That is, either the variance in each layer diverges, or the homogeneous Markov chain eventually "forgets" about its input and concentrates on "unlearnable functions". This was also pointed out by [1] in deep GPs, and corrected by adding skip connections, though directly to the input. All known NN architectures converge in the wide limit to their NN-GP kernel [2], and that the expression for the NTK is very similar to it. We know that the gradients for a wide feed-forward network converge to what would be predicted with the NTK [14, section 14], and that because of this similarity, training converges [14]. It is thus reasonable to expect that the NTK would describe the convergence for a trained ResNet very well, and also the gradients' variance should be well described by it (Theorem 4 of this work). Furthermore, Neal and [1] describe how a FFNet's kernel will forget the input and thus concentrate on unlearnable functions. Then, [4] derive using mean-field analysis that gradients are preserved for infinitely deep ReLU residual networks, so the functions they encode cannot forget the input. Thus, infinitely deep ResNets are learnable, and infinitely deep FFNets are not (theorem 5 of this paper). [1] Duvenaud, D., Rippel, O., Adams, R., & Ghahramani, Z., Avoiding pathologies in very deep networks. [2] Yang, Greg. https://papers.nips.cc/paper/9186-wide-feedforward-or-recurrent-neural-networks-of-any-architecture-are-gaussian-processes [3] Arora, S., Du, S. S., Hu, W., Li, Z., Salakhutdinov, R., & Wang, R., On exact computation with an infinitely wide neural net. [4] Yang, G., & Schoenholz, S., Mean field residual networks: on the edge of chaos [14] Allen-Zhu, Z., Li, Y., & Song, Z., A convergence theory for deep learning via over-parameterization,

Correctness: I did not carefully check the proofs, but the theorems are unsurprising and the methodology seems superficially correct. The empirical methodology is sound. I am not convinced of the statement in line 160: "Theorem 2 implies that a sufficiently wide FFNet trained by gradient flow is similar to the kernel regression predictor via its NTK". Why? Doesn't the kernel change from the initialization during the gradient flow? The proof of this is highly non-trivial and given by [14]. [14] Allen-Zhu, Z., Li, Y., & Song, Z., A convergence theory for deep learning via over-parameterization,

Clarity: It's fine. The paper is somewhat dense and hard to follow, but this is hard to avoid because it is very theoretical. Perhaps try to hold the reader's hand a bit more in section 3? The introduction has clearly been written hastily, and has several typos: L19 mechanism[s] L64 understand [the] generalization L67 essen[c]e L68 ha[s] been shown

Relation to Prior Work: I have complained at length about this in the "weaknesses" section.

Reproducibility: Yes

Additional Feedback:

[Author Response · NeurIPS 2020]

**Re: Reviewer #1**   For Thm. 4, we randomly initialize the ResNet with width=500, scaling factor $\gamma = 1$ and depth $L = 5, 10, 100, 300$, and then calculate the inner product of the Jacobians of the ResNet for two different inputs as in the definition of NTK (line 185-line187). We repeat the procedure for 500 times and plot the means of the empirical NTKs and the theoretical values in Fig. 1a which shows the two results match very well. For Thm. 5 and Thm. 6, Fig. 1b and Fig. 1c show that $\lim_{L\to\infty} |\overline{\Omega}_L(x,\tilde{x}) - 1/4| \cdot L/\log(L) \approx$ constant and $\lim_{L\to\infty} |\overline{\Omega}_L(x,\tilde{x}) - \overline{\Omega}_1(x,\tilde{x})| \cdot L \approx$ constant with $x^\top \tilde{x} = K_0$ chosen at 9 points.

(a) Thm. 4. $m = 500$ and scaling $\gamma = 1$

(b) Thm. 5. $y$-axis is $|\overline{\Omega}_L(x,\tilde{x}) - 1/4| \cdot L/\log(\text{L})$.

(c) Thm. 6. $y$-axis is $|\overline{\Omega}_L(x,\tilde{x}) - \overline{\Omega}_1(x,\tilde{x})| \cdot L$.

About ResNets' using convolutional layers (Conv) and batch norms(BN) in practice: we acknowledge that Conv and BN are indeed important and our techniques can be applied to them with some modification. We will add some discussions and leave them for our future investigation.

About the scaling factor: The scaling of the bottleneck weight effectively controls the norm of the activations of ResNets, and is commonly adopted in theoretical papers, e.g. [14] in our paper. In practice, the norm control is usually achieved by normalization techniques (batch norm, group norm, etc.).

**Re: Reviewer #2**   Thank you for the positive comments. About assumptions: we provide non-asymptotic results which bound the error between the finite-width NNs and the infinite-width NNs. Specifically, for ResNets with $\gamma = 1$, our theorem ensures good error control, where the width only depends logarithmically on the depth.

About generalizability: considering kernel ridge regression, one can show that the generalization error is "continuous" w.r.t. the kernel function under some integrable conditions. In this sense, the equivalence (in the view of NTK) means that the generalization of NTKs of sufficiently deep ResNets is close to the generalization of the NTK of 1-layer ResNet. The same applies to the poor generalization of deep FFNets.

**Re: Reviewer #3.**   About our motivation: our increasing depth analysis is a common practice in theoretical research. This is because the infinite-depth behavior is very similar to the large-depth behavior of NNs. This is analogous to we using central limit theorem in practice, but CLT essentially characterizes the limiting distribution based on infinite many samples. Moreover, we highlight that existing results are not limited to 50 and 100 layers. For example, [10] (He et al. 2016) even show that an ultradeep ResNets of 1001 layers can still generalize very well.

We remind that these ultradeep ResNets are not often used in practice because of their massive sizes. If implementable, their generalization performance can be better than existing SOTA results. As more advanced computational hardware (faster GPU's) is developed, deeper ResNets will become more popular in practice.

About our experiments: some existing results on NTK-based methods aim at achieving better performance. Therefore, their experiments use more complex structures with additional tricks, e.g. NTKs of ConvNets with Global Average Pooling. However, the goal of our experiments is to justify the correctness of our theorems – the NTKs of deep FFNets do not generalize, while the NTKs of deep ResNets generalize well. Therefore, the results of our experiments sufficiently serve the purpose of our paper.

About degrading kernel: the degeneration of NTKs of deep FFNets is clearly and rigorously stated in line 237-line 250.

About missing references: We will discuss them in the future version, but please kindly notice that Yang's paper is made public **after** the submission deadline of NeurIPS 2020.

**Re: Reviewer #4.**   Thank you for the positive comments and the references. The previous literatures are significant but have different aspects from ours. We will add more discussions in the next version.

About line 160: We are sorry for the misunderstanding here. Indeed, showing the equivalence of NNs trained by gradient flow and NTK kernel predictor is non-trivial. This result is rigorously proved in [22]. Thank you a lot for pointing out this issue. We will correct the statement in the next version.

[Meta-Review · NeurIPS 2020]

After the thorough discussion among the reviewers, there is a consensus that this is a good paper that warrants acceptance. There were some skepticisms in the initial reviews, but the authors have provided a rebuttal which addressed most of the major concerns. The reviewers have updated their reviews/scores accordingly. Hence, the paper is accepted as a poster. Based on my own judgement, the presentation of this paper should be improved in the camera-ready version. Moreover, please, include (at least in the appendix) the additional experiment result that you did to verify/proof-check your theorems. On top of that, the reviewers have already summarized their suggestion for changes that will help improve the paper further. Please, take them into consideration when preparing the camera-ready version of the paper.